# Riemann Tensor Neural Networks: Learning Conservative Systems with Physics-Constrained Networks

Anas Jnini [1]  Lorenzo Breschi [1]  Flavio Vella [1]

## Abstract

Divergence-free symmetric tensors (DFSTs) are fundamental in continuum mechanics, encoding conservation laws such as mass and momentum conservation. We introduce Riemann Tensor Neural Networks (RTNNs), a novel neural architecture that inherently satisfies the DFST condition to machine precision, providing a strong inductive bias for enforcing these conservation laws. We prove that RTNNs can approximate any sufficiently smooth DFST with arbitrary precision and demonstrate their effectiveness as surrogates for conservative PDEs, achieving improved accuracy across benchmarks. This work is the first to use DFSTs as an inductive bias in neural PDE surrogates and to explicitly enforce the conservation of both mass and momentum within a physics-constrained neural architecture.

## 1. Introduction

**Partial Differential Equations (PDEs)**  Partial Differential Equations (PDEs) are central to the mathematical modeling of complex physical systems, including fluid dynamics, thermodynamics, and material sciences. Traditional numerical methods, such as finite element and spectral methods, often require fine discretization of the physical domain to achieve high accuracy. These approaches can become computationally expensive, particularly in engineering applications where systems must be solved repeatedly under varying parameters or initial conditions. Recent advances in machine learning (ML) have shown promise in addressing these challenges by leveraging neural networks (NNs) as potential alternatives or enhancements to traditional numerical solvers (Kovachki et al., 2021; Li et al., 2020).

**Physical Inductive Biases in Machine Learning.**  A central limitation of generic neural models is their lack of built-in physical intuition. While convolutional or attention-based layers successfully exploit certain data symmetries (e.g., translation invariance), they do not automatically enforce fundamental physics, such as mass conservation or energy preservation. Physics-Informed Neural Networks (PINNs) (Lagaris et al., 1998; Raissi et al., 2019b; Cai et al., 2021; Haghighat et al., 2021; Hu et al., 2023) address this gap by adding PDE residuals and boundary conditions as soft constraints in the loss function. PINNs have been successfully deployed on many PDE problems (Karniadakis et al., 2021; Jnini et al., 2024b), but the "soft penalty" approach can lead to suboptimal enforcement of conservation laws and stiff optimization (Wang et al., 2021). Consequently, there is growing interest in *hard* or *explicit* constraints that guarantee PDE structure *a priori* (Richter-Powell et al., 2022; Greydanus et al., 2019b; Cranmer et al., 2020; Jnini et al., 2024a; Liu et al., 2024).

**Divergence-Free Symmetric Tensors in Physics and Mathematics.**  Divergence-free symmetric tensors (DFSTs) are a special class of tensor fields characterized by vanishing row-wise divergence and inherent symmetry in their indices. In an $(n + 1)$-dimensional space-time domain, where $n$ represents the spatial dimensions and the additional dimension accounts for time, these tensors frequently arise as stress or momentum flux tensors in various fields, including fluid dynamics, elasticity, kinetic theory, and relativistic hydrodynamics (Serre, 2018). For instance, the Navier–Stokes stress tensor or the compressible flux matrix can be represented as an $(n + 1) \times (n + 1)$ tensor $\mathcal{S}$ satisfying $\nabla \cdot \mathcal{S} = 0$. This formulation unifies the conservation of mass and momentum under a single flux-divergence constraint. Despite their natural alignment with PDE-based conservation laws, existing studies have not explored leveraging DFST structures *directly* as a neural inductive bias within neural network architectures.

**Our Contributions.**  In this work, we present a novel approach to embedding fundamental conservation laws directly into neural network architectures through DFSTs . Our primary contributions are as follows:

[1] Department of Information Engineering and Computer Science, University of Trento, Trento, Italy. Correspondence to: Anas Jnini <anas.jnini@unitn.it>.

*Proceedings of the 42$^{nd}$ International Conference on Machine Learning*, Vancouver, Canada. PMLR 267, 2025.

1. **Architectural Design of Riemann Tensor Neural Networks (RTNNs) :** We introduce RTNNs , a class of neural architectures specifically designed to generate DFST fields. RTNNs are tailored for approximating individual DFSTs, ensuring the divergence-free condition, $\nabla \cdot \mathcal{S} = 0$, is satisfied to machine precision.

2. **Theoretical Guarantees for RTNNs:** We establish theoretical foundations for RTNNs by proving their universal approximation capabilities for any sufficiently smooth DFST.

3. **Empirical Validation and Comparative Analysis:** We reformulate several benchmark problems within the DFST framework and conduct numerical experiments that demonstrate RTNNs consistently improve performance of PINNs in accuracy when used as surrogate models for conservative PDEs.

In the following sections, we review DFST-based PDE formulations, describe the proposed neural architectures, and present experimental validations on benchmark problems.

### 1.1. Related works

**Divergence-Free Symmetric Tensors in Mathematical Physics.** A large body of work by (Serre, 1997; 2018; 2019; 2021) has established the fundamental importance of DFSTs in continuum mechanics and kinetic theory. These tensors encode conservation principles for mass and momentum (in classical fluid dynamics) or energy–momentum (in relativistic hydrodynamics), and are present in models ranging from Euler or Boltzmann equations to mean-field (Vlasov–Poisson) descriptions of plasmas and galaxies. Although DFSTs have been investigated in PDE theory, prior investigations have largely focused on analytic or qualitative properties . To the best of our knowledge, no existing work leverages DFSTs explicitly as a numerical method or as an architectural inductive bias in machine learning frameworks.

**Hard-Constraints in Scientific Machine Learning.** Beyond the classical physics-informed approach of adding PDE residuals as soft constraints in the loss (Raissi et al., 2019a; Karniadakis et al., 2021), there is growing interest in incorporating *hard* constraints or specialized structures into neural networks. For instance, (Hendriks et al., 2020) investigate linearly constrained networks, (Richter-Powell et al., 2022) impose continuity-equation constraints via divergence-free vector fields, and several recent methods aim to preserve energy or momentum (Greydanus et al., 2019a; Cranmer et al., 2020). These efforts reflect a broader push in machine learning to embed *domain-specific priors*, thereby improving stability and generalization (LeCun et al., 1998; Giles & Maxwell, 1987). Our work similarly encodes the PDE structure "at the network level" via DFST, which

ensures strict conservation and symmetry. To the best of our knowledge, this work is the first to use enforce the conservation of both mass and momentum at the architectural level for surrogate modeling.

## 2. Background and Theory

**Notation (Preliminaries)** Let $n$ denote the spatial dimension and $\Omega \subset \mathbb{R}^n$ the spatial domain. The space-time domain is $\Omega_T = [0, T] \times \Omega \subset \mathbb{R}^{n+1}$, where $t \in [0, T]$ and $\mathbf{x} \in \Omega$. For a function $f(t, \mathbf{x})$, the augmented gradient is $\nabla f = (\partial_t f, \partial_{x_1} f, \ldots, \partial_{x_n} f)$, and the spatial gradient is $\widetilde{\nabla} f = (\partial_{x_1} f, \ldots, \partial_{x_n} f)$.

**Divergence-Free Symmetric Tensors in Continuum Mechanics.** We begin by introducing the class of *divergence-free symmetric tensors*, which encode either the conservation of mass and momentum in classical mechanics or energy and momentum in special relativity. A tensor field

$$S : \Omega_T \to \mathbb{R}^{(n+1) \times (n+1)}$$

is said to be *symmetric* if $S = S^\top$, and *divergence-free* if it satisfies

$$\text{Div}_{t,x}(S)_i := \partial_t s_{i0} + \sum_{j=1}^n \partial_{x_j} s_{ij} = 0, \quad \forall i \in \{0, \ldots, n\}.$$

Many classical PDE systems, including compressible or incompressible fluid flow, elasticity, and shallow-water models, can be expressed in this flux-divergence form by suitably choosing $S$. A canonical representation in fluid mechanics is:

$$S = \begin{pmatrix} \rho & m^\top \\ m & \dfrac{m \otimes m}{\rho} + \sigma \end{pmatrix}, \quad (1)$$

where $\rho : \Omega_T \to \mathbb{R}$ denotes the mass density, $m : \Omega_T \to \mathbb{R}^n$ the linear momentum field, and $\sigma : \Omega_T \to \mathbb{R}^{n \times n}$ the stress tensor. Enforcing $\text{Div}_{t,x}(S) = 0$ then unifies mass and momentum conservation, while additional constraints (e.g., constitutive laws or energy equations) can further specify $\sigma$ or couple $\rho$ and $m$.

**Motivation for Neural Parametrization.** Although one can penalize the residuals of the condition $\text{Div}_{t,x}(S) = 0$ in a soft-constraint manner (e.g., through terms in the loss function), this approach does not guarantee the satisfaction of the divergence-free condition, especially when optimization is challenging or regularization terms are underweighted; this is often the case in the non-linear regime that we are considering, as described in (Bonfanti et al., 2024).

Instead, we propose embedding the divergence-free property directly into the neural network architecture. A primary motivation for this architectural choice, beyond the

immediate benefit of ensuring strict conservation, is to more fundamentally address how coupled physical quantities are represented. In many physical systems, such as those in fluid dynamics, variables like density, velocity, and pressure are not independent but are deeply interrelated. Conventional neural network models that treat these as separate output channels can struggle to capture these intrinsic physical and geometric correlations—for instance, the interdependence between velocity components or the unified response of density and velocity to pressure gradients. By designing our network to output a DFST, we aim to inherently model these quantities as components of a single, unified field, linked directly by the underlying conservation laws. Consequently, such a hard-coded constraint ensures strict conservation (e.g., of mass and momentum) to machine precision, while also providing better physical consistency and a stronger inductive bias.

**Proposed Approach.** The following sections introduce a neural-network-based construction that guarantee the output is a divergence-free symmetric tensor. We prove that our approach can approximate any sufficiently smooth DFST to arbitrary accuracy, thus offering a robust way to integrate conservation principles into neural PDE solvers.

## 2.1. Constructing Divergence-Free Symmetric Tensors on a Flat Manifold (DFSTs)

**Theorem 2.1** (Representation of Divergence-Free Symmetric Tensors on a Flat Manifold). *Let $V$ be an $n$-dimensional real vector space with a fixed basis $\{e_a\}_{a=1}^n$, and let $\{e_a^*\}_{a=1}^n$ denote the corresponding dual basis of $V^*$. Let $\Lambda^2 V^*$ denote the space of 2-forms on $V$. Consider the space of all $(0,4)$-tensors $K_{abcd}$ defined on a flat manifold equipped with a Levi-Civita connection $\nabla$, satisfying the following symmetries:*

*1. Antisymmetry within index pairs:*

$$K_{(ab)cd} = 0, \tag{2}$$
$$K_{ab(cd)} = 0, \tag{3}$$

*2. Symmetry between pairs:*

$$K_{abcd} = K_{cdab}. \tag{4}$$

*Let $\{\omega_1, \ldots, \omega_m\}$ be a fixed basis of $\Lambda^2 V^*$, where $m = \frac{n(n-1)}{2}$. Define the tensors:*

$$T_{abcd}^{(i,j)} := \omega_i(e_a^* \wedge e_b^*)\omega_j(e_c^* \wedge e_d^*) + \omega_j(e_a^* \wedge e_b^*)\omega_i(e_c^* \wedge e_d^*), \tag{5}$$

*where $e_a^* \wedge e_b^*$ is the wedge product of the dual basis elements $e_a^*$ and $e_b^*$.*

*The space of divergence-free symmetric tensors $S_{ab}$ on the flat manifold is the image of the map:*

$$S_{ab} = \nabla^c \nabla^d K_{acbd}, \tag{6}$$

*where $K_{abcd}$ is any $(0,4)$-tensor satisfying the symmetries (2)–(4). Any $S_{ab}$ can be expressed as:*

$$S_{ab} = \sum_{1 \le i \le j \le m} T_{acbd}^{(i,j)} \nabla^c \nabla^d c_{ij}, \tag{7}$$

*where $c_{ij}$ are smooth scalar functions.*

*Proof.* The proof is presented in Appendix A. □

### 2.2. Riemann Tensor Neural Network

In the setting where we wish to approximate a divergence-free symmetric tensor field

$$S \colon \Omega \to \mathbb{R}^{n \times n}, \tag{8}$$

we define a **Riemann Tensor Neural Network** as follows.

**Definition 2.2** (Riemann Tensor Neural Network[1]). Suppose:

- $\Omega \subset \mathbb{R}^n$ is a flat domain (manifold without boundary or with suitable BCs),

- $\{T_{abcd}^{(i,j)}\}$ is a finite *non-trainable* basis of Riemann-like $(0,4)$-tensors as defined in Theorem 2.1,

- $\mathrm{NN}_\theta \colon \mathbb{R}^k \to \mathbb{R}^{\frac{m(m+1)}{2}}$ is a multilayer perceptron (MLP) with twice-differentiable activations, whose input $x \in \mathbb{R}^k$ indexes points in $\Omega$.

Then an *RTNN* for the single-field case is constructed by:

1. **Scalar coefficients** $\{c_{ij}(x;\theta)\}$: The MLP $\mathrm{NN}_\theta$ outputs $\frac{m(m+1)}{2}$ scalar functions $\{c_{ij}(x;\theta)\}_{1 \le i \le j \le m}$.

2. **Hessian Computation:** For each $c_{ij}(x;\theta)$, compute the Hessian components $\partial^c \partial^d c_{ij}(x;\theta)$ via automatic differentiation.

3. **Tensor Field Construction:** Define the $(0,2)$-tensor field

$$S_\theta(x) = \sum_{1 \le i \le j \le m} T_{acbd}^{(i,j)} \partial^c \partial^d c_{ij}(x;\theta). \tag{9}$$

By design, $S_\theta(x)$ is row-wise divergence-free and symmetric for all $x \in \Omega$.

We call $S_\theta$ a **Riemann Tensor Neural Network**. It provides a parametric approximation $S_\theta \approx S$ to a single DFST on $\Omega$.

*Remark* 2.3. Although the network outputs scale as $m(m-1)$, the problem setup is generally overparameterized. Depending on the application, certain scalar functions can be set to zero without violating the DSFT condition, provided that the number of basis functions exceeds the degrees of freedom.

### 2.3. Universal Approximation Theorem for RTNN

**Theorem 2.4** (Universal Approximation for RTNN). *Let* $\Omega \subset \mathbb{R}^n$ *be a bounded (hence compact) domain, and suppose* $S \colon \Omega \to \mathbb{R}^{n \times n}$ *is a* $\mathbf{C}^2$*-smooth, divergence-free, symmetric tensor field on* $\Omega$*. Then for any* $\varepsilon > 0$*, there exists a* Riemann Tensor Neural Network $S_\theta$ *such that*

$$\sup_{x \in \Omega} \|S(x) - S_\theta(x)\|_{\mathrm{Fro}} < \varepsilon, \tag{10}$$

*where* $\| \cdot \|_{\mathrm{Fro}}$ *denotes the Frobenius norm on* $\mathbb{R}^{n \times n}$*. In particular,* $S_\theta$ *also remains divergence-free and symmetric on* $\Omega$*.*

*Proof.* We present the proof in Appendix A.3 □

## 3. Methodology and Applications

The preceding sections established the theoretical foundation of divergence-free symmetric tensors DFSTs and RTNNs as a rigorous approach to enforcing conservation laws within neural architectures. In the following section, we move from theory to application, showcasing how RTNNs can be employed to model various conservative systems, including the Euler and Navier-Stokes equations, and Magneto-Hydrodynamics(MHD).

### 3.1. Efficient Implementation and Practical Considerations

**Automatic Differentiation** We employ Taylor-Mode Automatic Differentiation, which propagates Taylor coefficients through the network by treating the computational graph as an augmented network with weight sharing. This approach effectively reduces redundant computations associated with higher-order derivatives, significantly accelerating the training process of RTNNs. Additionally, for Magneto-Hydrodynamics in Section 3.4, we utilize Separable Physics-Informed Neural Networks (SPINNs). SPINNs decompose PDE residuals into per-axis evaluations, facilitating efficient differential operations on large-scale regular grids (Cho et al., 2023).

**Optimization and Stability** Our method models densities and momenta instead of velocity fields, velocity recov-

---

[1]So named because the underlying $(0,4)$-tensors share index symmetries with the Riemann curvature tensor in differential geometry.

ery involves dividing by $\rho$, leading to instability when $\rho$ is initialized around a small value (Richter-Powell et al., 2022). To address this, we add an identity matrix to $\mathcal{S}_\theta$, ensuring $\rho$ is initialized near 1 without violating DFST constraints. Throughout the experiments in this paper, we use the Least-memory BFGS (Nocedal & Wright, 1999) optimizer due to the highly non-linear nature of our problems. Additionally, LBFGS efficiently approximates second-order curvature information, facilitating effective optimization in the complex, non-linear loss landscapes encountered in our experiments. The challenges of optimizing in such regimes and the importance of second-order optimizers have been well documented in the literature (Jnini et al., 2024b; Müller & Zeinhofer, 2024; Bonfanti et al., 2024).

**Code implementation and public repository** Our code has been implemented using the JAX library (Bradbury et al., 2018). Our implementation is publicly available at https://github.com/HicrestLaboratory/Riemann-Tensor-Neural-Networks.

### 3.2. Pedagogic Example: 2D Isentropic Euler Vortex

For this pedagogic example, we simulate a 2D isentropic Euler vortex—a smooth, rotational flow solution to the Euler equations that accurately captures vortex dynamics and is commonly used as a standard benchmark for evaluating the accuracy of numerical solvers—over a spatial domain $\Omega = [0, L_x] \times [0, L_y]$ and a time interval $[0, T]$. We employ well-defined analytical initial and boundary conditions that are detailed in Appendix B.1.

Let $\rho > 0$ denote the density, $(u, v)$ the velocity field, and $p$ the pressure. For isentropic flow with $\gamma > 1$, the pressure is given by $p(\rho) = \kappa \rho^\gamma$. The governing equations, including the energy equation, are:

**Governing Equations** The 2D compressible Euler equations are:

$$\partial_t \rho + \tilde{\nabla} \cdot (\rho u, \rho v) = 0, \tag{11}$$

$$\partial_t(\rho u) + \tilde{\nabla} \cdot (\rho u^2, \rho u v) = -\partial_x p, \tag{12}$$

$$\partial_t(\rho v) + \tilde{\nabla} \cdot (\rho u v, \rho v^2) = -\partial_y p, \tag{13}$$

$$\partial_t E + \tilde{\nabla} \cdot ((E + p)u, (E + p)v) = 0, \tag{14}$$

where $E = \frac{p}{\gamma - 1} + \frac{1}{2}\rho(u^2 + v^2)$ is the total energy.

**DFST Formulation and Decomposition of** $\sigma$**.** Rewriting (11)–(13) in flux-divergence form, we express the system as:

$$\nabla_{t,x,y} \cdot \mathcal{S} = 0, \quad \text{where}$$

$$\mathcal{S} = \begin{pmatrix} \rho & \rho u & \rho v \\ \rho u & \frac{(\rho u)^2}{\rho} + \sigma_{xx} & \frac{(\rho u)(\rho v)}{\rho} \\ \rho v & \frac{(\rho v)(\rho u)}{\rho} & \frac{(\rho v)^2}{\rho} + \sigma_{yy} \end{pmatrix}.$$

Here, the stress tensor $\sigma$ is:

$$\sigma = p\mathbf{I} + \underbrace{0}_{\text{deviatoric part}},$$

where $p = p(\rho)$ represents the isotropic pressure contribution. The absence of a deviatoric term reflects the assumption of inviscid flow. The divergence-free condition $\nabla \cdot \mathcal{S} = 0$ enforces:

- mass conservation, and

- momentum conservation in $x$- and $y$-directions.

Additionally, the energy equation is given separately as:

$$\partial_t E + \tilde{\nabla} \cdot \Big( (E+p)u, (E+p)v \Big) = 0,$$

where the total energy $E$ is:

$$E = \frac{p}{\gamma - 1} + \frac{1}{2}\rho(u^2 + v^2).$$

**RTNN Parametrization.** To approximate solutions of (11)–(14), we define a family of tensors $\mathcal{S}_\theta$ using RTNNs . The parametrization proceeds as follows:

1. **RTNN Parametrization of $\mathcal{S}_\theta$:** Let $\mathcal{S}_\theta$ denote an RTNN as described in Section 2.2. By construction, $\mathcal{S}_\theta$ is *divergence-free and symmetric* in $\Omega_T$.

2. **Extracting Physical Fields:** We can interpret $\mathcal{S}_\theta$ in block form. From it, we read off::

$$\rho_\theta = (\mathcal{S}_\theta)_{0,0}, \quad (\rho_\theta u_\theta, \rho_\theta v_\theta) = (\mathcal{S}_\theta)_{1:2,0},$$

$$\sigma_\theta = (\mathcal{S}_\theta)_{1:2,1:2} - \frac{(\rho_\theta u_\theta, \rho_\theta v_\theta) \otimes (\rho_\theta u_\theta, \rho_\theta v_\theta)}{\rho_\theta}.$$

3. **Zero-Deviatoric Constraint and Energy Parametrization:** We can parametrize the pressure by decomposing the stress tensor into isotropic and deviatoric parts:

$$\sigma_\theta = p_\theta \mathbf{I} + \sigma_\theta^{\text{dev}},$$

where:

$$p_\theta = \frac{1}{2}\text{tr}(\sigma_\theta), \qquad \sigma_\theta^{\text{dev}} = \sigma_\theta - p_\theta \mathbf{I}. \qquad (15)$$

Additionally, we parametrize the energy as:

$$E_\theta = \frac{p_\theta}{\gamma - 1} + \frac{1}{2}\rho_\theta(u_\theta^2 + v_\theta^2).$$

For an inviscid isentropic vortex, we enforce the constraint $\sigma_\theta^{\text{dev}} = 0$ during training.

While the parametrized fields exactly satisfy the DSFT constraints, they are only solutions to the momentum equations if the stress tensor satisfies the zero deviatoric constraints, which we can penalize in the loss function in addition to the boundary and initial terms.

We clarify the equivalence of the zero deviatoric constraint with the momentum residual in Appendix Section A.4.

**Loss Function.** To train the RTNN and ensure that the modeled tensor $\mathcal{S}_\theta$ adheres to the governing equations and boundary conditions, we define an objective function:

$$\mathcal{L}(\theta) = \mathcal{L}_{\text{BC}} + \mathcal{L}_{\text{IC}} + \mathcal{L}_\sigma + \mathcal{L}_E,$$

where $\mathcal{L}_{\text{BC}}$ penalizes deviations from the prescribed boundary conditions while $\mathcal{L}_{\text{IC}}$ enforces consistency with initial conditions. The term $\mathcal{L}_\sigma$ ensures that the stress tensor remains purely isotropic by penalizing the magnitude of the deviatoric component, $\|\sigma_\theta^{\text{dev}}\|^2$, it's inclusion is equivalent to penalizing the momentum equation residual as shown in in A.4. Finally, $\mathcal{L}_E$ minimizes the residual of the energy equation, measured as $\|\partial_t E_\theta + \tilde{\nabla} \cdot ((E_\theta + p_\theta)(u_\theta, v_\theta))\|^2$, ensuring that the total energy is properly conserved within the system. All loss terms are formulated in the least squares sense.

**Experimental Setup.** For the neural network training, we sample **500 interior collocation points** within $\Omega \times [0, T]$ to enforce the residual constraints of the governing PDE. Additionally, **100 boundary and initial condition points** are sampled to impose the prescribed constraints. Our RTNN model is parameterized by a **Multilayer Perceptron (MLP)** with **4 hidden layers**, each containing **50 neurons**. Training is performed entirely **without labeled data**, the model is validated against the analytical solution of the isentropic Euler vortex to evaluate accuracy.

We benchmark RTNN against two methods: (1) the standard PINN approach and (2) Neural Conservation Laws (NCL) that enforces exact mass conservation(Richter-Powell et al., 2022). Both methods use similar MLP architectures for fairness. Performance is evaluated in terms of median average relative $L^2$ error on all fields and simulation time. We train all three methods using 200,000 iterations of the L-BFGS optimizer. We follow the loss scheme presented in this section, while training both PINN and NCL using PDE residuals penalized in the loss.

**Results and Discussion.** Table 1 summarizes the results, while Figure 1 presents the evolution of the relative $L^2$ error over simulation time. Our RTNN significantly outperforms both PINN and NCL, achieving a median relative $L^2$ error that is two orders of magnitude lower than PINN and four orders of magnitude lower than NCL. Furthermore,

| Method | Relative $L_2$ Error | Wall Time (s) |
|--------|----------------------|---------------|
| **RTNN** | **9.92e-05** | 596.34 |
| NCL | 3.87e-01 | 2008.57 |
| PINN | 3.82e-02 | **365.67** |

*Table 1.* Comparison of methods for the Euler experiment, reporting the median relative $L_2$ error and median wall time across five independent training runs with different random seeds.

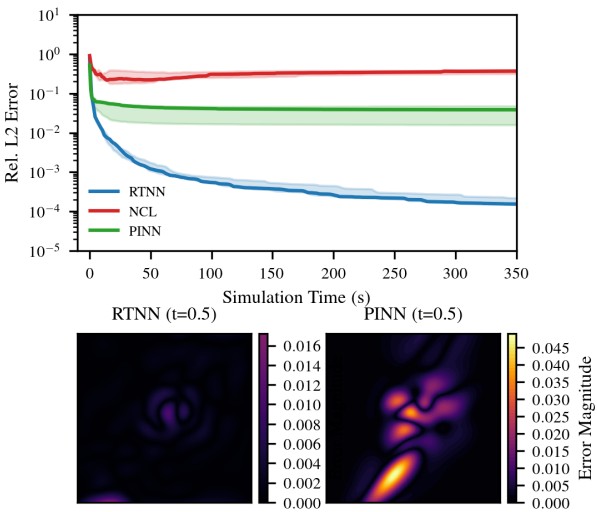

*Figure 1.* Comparison of training dynamics and error fields for RTNN, NCL, and PINN for the Euler experiment. **Top**: relative $L^2$ error evolution over simulation time, showing median (lines) and IQR (shaded). **Bottom**: Error fields for RTNN (left) and PINN (right) at $t = 0.5$

RTNN demonstrates stable convergence while maintaining competitive training times.

### 3.3. Incompressible Navier-Stokes Equation

We next consider the incompressible Navier-Stokes equations, which include additional viscous forces compared to inviscid Euler flows.

**Governing Equations.** The incompressible Navier-Stokes system in the $n$-dimensional case is given by:

$$\tilde{\nabla} \cdot \mathbf{u} = 0, \tag{16}$$

$$\partial_t \mathbf{u} + (\mathbf{u} \cdot \tilde{\nabla})\mathbf{u} = -\tilde{\nabla}p + \nu\Delta\mathbf{u}, \tag{17}$$

where $\mathbf{u} = (u, v, w)$ represents the velocity field, $p$ denotes the pressure field, and $\nu > 0$ is the kinematic viscosity. Equation (16) enforces the incompressibility condition, ensuring that the divergence of the velocity field is zero. Mean-

while, equation (17) balances the convective, pressure, and viscous forces within the fluid. The term $\nu\Delta\mathbf{u}$ specifically models the internal fluid friction due to viscosity.

**DFST Formulation and Stress Decomposition.** These equations can be expressed in a divergence-free symmetric tensor (DFST) form. We define a tensor

$$\mathcal{S} = \begin{pmatrix} 1 & \mathbf{u}^\top \\ \mathbf{u} & \mathbf{u} \otimes \mathbf{u} + \sigma \end{pmatrix},$$

such that

$$\nabla \cdot \mathcal{S} = 0.$$

Here, the term $\mathbf{u} \otimes \mathbf{u}$ represents the convective flux, while $\sigma$ is the total stress tensor decomposed as

$$\sigma = p\mathbf{I} + \sigma^{\mathrm{dev}},$$

where the deviatoric part $\sigma^{\mathrm{dev}}$ captures viscous stresses via

$$\sigma^{\mathrm{dev}} = \nu\big(\tilde{\nabla}\mathbf{u} + (\tilde{\nabla}\mathbf{u})^\top\big).$$

**Exact incompressibility.** To enforce exact incompressibility (i.e., $S_{00} = 1$), observe that any contributions to $S_{00}$ come specifically from basis 2-forms containing $e_0^*$. Consequently, by choosing the corresponding coefficients $c_{ij}$ to vanish whenever the wedge product includes $e_0^*$ in $T_{abcd}$, we ensure that $S_{00}$ consists only of the identity term we added for stability, thus achieving exact incompressibility.

**RTNN Parametrization.** Following Section 3.2, we employ an RTNN $\mathcal{S}_\theta$ to represent the solution. From its block structure, we extract the physical fields as:

$$\rho_\theta = 1, \quad (u_\theta, v_\theta, w_\theta) = (\mathcal{S}_\theta)_{1:n,0},$$

$$\sigma_\theta = (\mathcal{S}_\theta)_{1:n,1:n} - (u_\theta, v_\theta, w_\theta) \otimes (u_\theta, v_\theta, w_\theta).$$

Here, $\rho_\theta$ represents the density field, while $(u_\theta, v_\theta, w_\theta)$ correspond to the velocity components. The viscous stress tensor $\mathcal{D}_\theta$ can be computed via automatic differentiation applied directly to the velocity field. Instead, we define:

$$\mathcal{D}_\theta^{\mathrm{dev}} = \nu\left(\tilde{\nabla}\mathbf{u}_\theta + (\tilde{\nabla}\mathbf{u}_\theta)^\top\right).$$

Although this introduces additional computational complexity, it can be mitigated using Taylor-mode automatic differentiation. The stress tensor can be decomposed into an isotropic part and a deviatoric part:

$$\sigma_\theta = p_\theta\mathbf{I} + \sigma_\theta^{\mathrm{dev}}, \quad p_\theta = \frac{1}{n}\mathrm{tr}(\sigma_\theta).$$

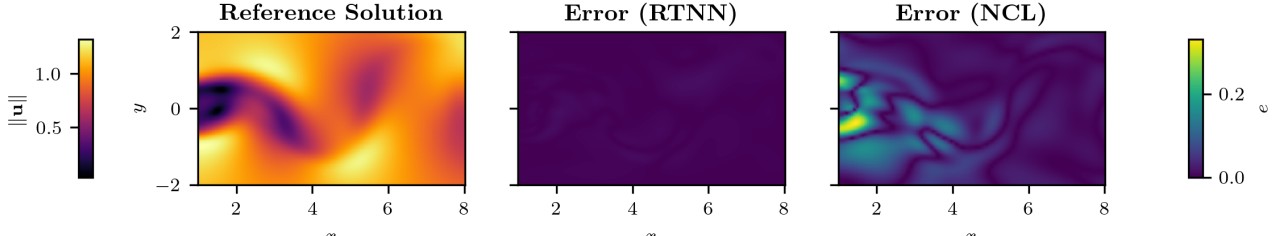

*Figure 2.* Qualitative comparison of predicted velocity fields for the unsteady Cylinder Wake flow at $Re = 100$ and a representative time instance $t = 7.0$. Left: Reference solution showing the reference velocity magnitude $\|\|\mathbf{u}\|\|$. Middle: Absolute error of the RTNN prediction, $e = \|\mathbf{u}_{\text{Ref}} - \mathbf{u}_{\text{RTNN}}\|_2$. Right: Absolute error of the NCL prediction, $e = \|\mathbf{u}_{\text{Ref}} - \mathbf{u}_{\text{NCL}}\|_2$. This visualization clearly demonstrates RTNN's superior accuracy in capturing the complex vortex shedding structures with significantly lower error compared to NCL.

**Viscous Residual and Loss Function.** To enforce momentum balance, we focus on matching the RTNN-derived deviatoric stress $\sigma_\theta^{\text{dev}}$ with the velocity-based viscous stress $\nu(\nabla \mathbf{u}_\theta + (\nabla \mathbf{u}_\theta)^\top)$. Define:

$$\mathcal{R}_{\sigma,\text{viscous}} = \sigma_\theta^{\text{dev}} - \mathcal{D}_\theta$$

Because pressure can act as a scalar offset in this formulation, ensuring correct deviatoric stresses is sufficient to satisfy the momentum equation. Consequently, our training objective can be written:

$$\mathcal{L}(\theta) = \left\|\mathcal{R}_{\sigma,\text{viscous}}\right\|_{\Omega_T}^2 + \mathcal{L}_{\text{BC}} + \mathcal{L}_{\text{IC}} + \mathcal{L}_{\text{data}},$$

where $\mathcal{L}_{\text{BC}}$ and $\mathcal{L}_{\text{IC}}$ enforce boundary and initial conditions, and $\mathcal{L}_{\text{data}}$ penalizes any available labeled measurements. All loss terms are formulated in the least squares sense.

**Experimental Setups.** We validate our approach on three representative incompressible Navier–Stokes scenarios:

- **3D Beltrami Flow** We consider the three-dimensional Beltrami flow at a Reynolds number of $Re = 1$ to verify the accuracy of our RTNN framework. The computational domain is discretized using **2,601 interior collocation points** to enforce the PDE residuals, supplemented by **961 boundary and initial condition points** to impose the necessary constraints. An MLP with **4 hidden layers** and **50 neurons per layer** employing Tanh activation functions is utilized. The model is trained using **100,000 iterations** of the L-BFGS optimizer without any labeled data. Validation is performed against **26,000 interior points** sampled within the domain to assess the model's performance. Detailed setup information and error plots are provided in Appendix B.2.

- **Steady Flow around a NACA Airfoil** This experiment investigates the steady laminar flow at $Re = 1000$ around a NACA 0012 airfoil. The steady-state problem is addressed by treating time as a dummy dimension set to zero in the forward pass. The computational domain is discretized with **40,000 collocation points** to enforce PDE residuals and boundary conditions, alongside **2,000 labeled data points** obtained from an in-house Reynolds-Averaged Navier-Stokes (RANS) solver to supervise the training. An MLP consisting of **4 hidden layers** and **50 neurons per layer** with Tanh activation functions is employed. The model undergoes **50,000 iterations** of the L-BFGS optimizer. Validation is conducted on the **14,000 collocation points** to evaluate accuracy. Further setup details and visual results are provided in Appendix B.4.

- **Cylinder Wake** We simulate a two-dimensional unsteady vortex-shedding flow at $Re = 100$ around a circular cylinder centered at $(0, 0)$. The computational domain is defined as $[1, 8] \times [-2, 2]$ with the time interval $[0, 7]$, discretized in increments of $\Delta t = 0.1$. The domain is discretized using **40,000 interior collocation points** to enforce the PDE residuals and **5,000 boundary and initial condition points** to apply the necessary constraints. A **4-layer, 50-neuron MLP** with Tanh activation functions is trained using **50,000 iterations** of the L-BFGS optimizer without any labeled data. Validation is performed using Direct Numerical Simulation (DNS) data from Raissi et al. (2019a) to assess the model's performance. Comprehensive setup details and error analyses are presented in Appendix B.3.

**Results and Discussion.** Table 3.3 summarize the performance of our RTNN formulation compared to NCL and standard PINNs by showing median average relative $L^2$ error across all fields and wall times, while Figures 4 and 6 summarize the training dynamics for these experiments and Figure 5 provides a qualitative comparison for the cylinder wake. As observed in the table, our ansatz delivers improved

| | Cylinder | | Airfoil | | Beltrami | |
|---|---|---|---|---|---|---|
| Method | rL2 Error | Time (s) | rL2 Error | Time (s) | rL2 Error | Time (s) |
| RTNN | **5.70e-03** | 1.21e+03 | **1.44e-02** | 1.10e+03 | **4.28e-04** | 2.97e+02 |
| NCL | 2.54e-02 | 2.46e+03 | 1.53e-01 | 2.39e+03 | 1.73e-03 | 1.00e+03 |
| PINN | 2.99e-02 | **3.12e+02** | 2.48e-01 | **1.06e+03** | 1.41e-03 | **1.82e+02** |

*Table 2.* Comparison of RTNN, NCL, and PINN across three Incompressible Navier-Stokes experiments (Cylinder, Airfoil, Beltrami). We report median relative $L^2$ (rL2) error and median wall times across 5 different seeds.

accuracy while achieving comparable training times. Across all three test cases, the RTNN approach consistently yields lower relative errors, indicating its potential for robust, data-efficient modeling of incompressible Navier–Stokes flows for both self-supervised learning in the PINN manner and in in scarce-data scenarios. Furthermore, for the cylinder wake, Figure 5 visually demonstrates RTNN's superior capability in capturing the intricate vortex shedding structures. While the relative $L^2$ error provides a global measure of accuracy, the qualitative results in the figure highlight that RTNN yields solutions that are particularly effective in the complex wake region, where capturing fine details is crucial, an aspect that can be averaged out in global error metrics.

### 3.4. Magnetohydrodynamics (MHD)

We next consider the incompressible resistive magnetohydrodynamics (MHD) equations, which couple fluid velocity and pressure to a magnetic field. The governing PDEs on a domain $\Omega \times [0, T]$ are:

$$\partial_t \mathbf{u} + (\mathbf{u} \cdot \tilde{\nabla})\,\mathbf{u} = -\tilde{\nabla}\Big(p + \tfrac{|\mathbf{B}|^2}{2}\Big) + (\mathbf{B} \cdot \tilde{\nabla})\mathbf{B} + \nu\,\tilde{\nabla}^2\mathbf{u}, \tag{18}$$

$$\partial_t \mathbf{B} + (\mathbf{u} \cdot \tilde{\nabla})\,\mathbf{B} = (\mathbf{B} \cdot \tilde{\nabla})\,\mathbf{u} + \eta\,\tilde{\nabla}^2\mathbf{B}, \tag{19}$$

$$\tilde{\nabla} \cdot \mathbf{u} = 0, \qquad \tilde{\nabla} \cdot \mathbf{B} = 0, \tag{20}$$

where $\mathbf{u} = (u, v)$ is the velocity field (2D case), $p$ is the pressure, $\mathbf{B} = (B_x, B_y)$ is the magnetic-field vector, $\nu$ is the kinematic viscosity, $\eta$ is the magnetic diffusivity, and $|\mathbf{B}|^2 = B_x^2 + B_y^2$.

**Vector-Potential Formulation of the Magnetic Field.** To enforce $\tilde{\nabla} \cdot \mathbf{B} = 0$ exactly, we parametrize $\mathbf{B}$ via a *vector potential* $\mathbf{A}$:

$$\mathbf{B} = \tilde{\nabla} \times \mathbf{A}.$$

In 2D, we may simply take $\mathbf{A} = (0, 0, \psi(x, y))$, so that $\mathbf{B} = (\partial_y \psi, -\partial_x \psi)$ automatically satisfies $\tilde{\nabla} \cdot \mathbf{B} = 0$. The induction equation (19) then becomes an evolution for $\psi$:

$$\partial_t \psi + (\mathbf{u} \cdot \tilde{\nabla}a)\,\psi = \eta\,\tilde{\nabla}^2\psi.$$

**Divergence-Free Symmetric Tensor (DFST) for Momentum.** Similar to the Navier–Stokes case, we unify incompressibility and momentum conservation in the DSFT Form:

$$\mathcal{S} = \begin{pmatrix} 1 & \mathbf{u}^\top \\ \mathbf{u} & \mathcal{T} \end{pmatrix}, \tag{21}$$

where $\mathbf{u} \in \mathbb{R}^2$ is:

$$\mathcal{T} = p\,\mathbf{I} + \mathbf{u} \otimes \mathbf{u} - \nu\big(\tilde{\nabla}\mathbf{u} + (\tilde{\nabla}\mathbf{u})^\top\big) + \mathcal{M}^{\mathrm{dev}},$$

with $\mathcal{M}^{\mathrm{dev}}$ the the Maxwell magnetic stress.

$$\mathcal{M}^{\mathrm{dev}} = \tfrac{1}{2}\,|\mathbf{B}|^2\,\mathbf{I} - \big(\mathbf{B} \otimes \mathbf{B}\big),$$

for magnetic field $\mathbf{B}$.

**RTNN Parametrization for 2D Incompressible MHD**
Let $\mathcal{S}_\theta$ be a $(2+1)\times(2+1)$ RTNN. From its block structure, we extract the physical fields as:

$$(u_\theta, v_\theta) = (\mathcal{S}_\theta)_{1:2,\,0}$$

We parametrize the magnetic field $\mathbf{B}_\theta$ via a separate network outputting a scalar potential $\psi_\theta$:

$$\mathbf{B}_\theta = \tilde{\nabla} \times (0, 0, \psi_\theta) = \big(\partial_y \psi_\theta, -\partial_x \psi_\theta\big), \quad \tilde{\nabla}\cdot\mathbf{B}_\theta = 0.$$

Define the Maxwell stress (including its isotropic part):

$$\mathcal{M}_\theta = \tfrac{1}{2}\,|\mathbf{B}_\theta|^2\,\mathbf{I} - \big(\mathbf{B}_\theta \otimes \mathbf{B}_\theta\big).$$

We then let

$$\sigma_\theta = (\mathcal{S}_\theta)_{1:2,\,1:2} - \big(u_\theta, v_\theta\big) \otimes \big(u_\theta, v_\theta\big) - \mathcal{M}_\theta,$$

so $\sigma_\theta$ is the purely *fluid* portion of the stress once advection and magnetic terms have been subtracted. We then proceed in the same manner as section 3.3.

$$\sigma_\theta = p_\theta\,\mathbf{I} + \sigma_\theta^{\mathrm{dev}}, \quad p_\theta = \tfrac{1}{2}\,\mathrm{tr}\big(\sigma_\theta\big),$$

and define the viscous stress term

$$\mathcal{D}_\theta = \nu\big(\tilde{\nabla}\mathbf{u}_\theta + (\tilde{\nabla}\mathbf{u}_\theta)^\top\big).$$

**Training Objective.** Enforcing momentum balance then requires $\sigma_\theta^{\mathrm{dev}} \approx \mathcal{D}_\theta^{\mathrm{dev}}$. We form a residual

$$\mathcal{R}_{\sigma,\mathrm{visc}} = \sigma_\theta^{\mathrm{dev}} - \mathcal{D}_\theta,$$

penalized in the loss. In addition, we include the induction equation residual

$$\mathcal{R}_{\mathrm{induction}} = \partial_t\mathbf{B}_\theta + (\mathbf{u}_\theta\tilde{\nabla})\,\mathbf{B}_\theta - (\mathbf{B}_\theta\tilde{\nabla})\,\mathbf{u}_\theta - \eta\,\tilde{\nabla}^2\mathbf{B}_\theta,$$

Leading to the final training objective:

$$\mathcal{L}(\theta) = \|\mathcal{R}_{\sigma,\mathrm{visc}}\|_{\Omega_T}^2 + \|\mathcal{R}_{\mathrm{induction}}\|_{\Omega_T}^2 + \mathcal{L}_{\mathrm{BC}} + \mathcal{L}_{\mathrm{IC}} +$$
$$\mathcal{L}_{\mathrm{data}}.$$

The first two terms penalize violations of momentum conservation (viscous + magnetic) and induction equations, respectively. $\mathcal{L}_{\mathrm{BC}}$ and $\mathcal{L}_{\mathrm{IC}}$ enforce boundary and initial conditions, and $\mathcal{L}_{\mathrm{data}}$ integrates available labeled observations. All loss terms are formulated in the least squares sense.

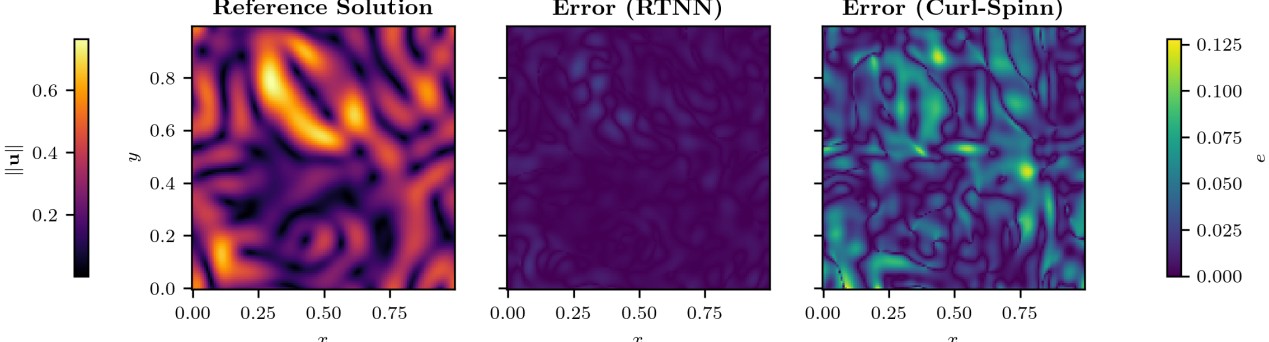

*Figure 3.* Qualitative comparison of predicted velocity fields for the 2D periodic incompressible MHD experiment ($Re = 1000$, $ReM = 1000$) at time $t = 0.55$. Left: Reference solution showing the velocity magnitude $\|\|\mathbf{u}\|\|$. Middle: Absolute error of the RTNN prediction, calculated as $e = \|\mathbf{u}_{\text{Ref}} - \mathbf{u}_{\text{RTNN}}\|_2$. Right: Absolute error of the SPINN baseline prediction, calculated as $e = \|\mathbf{u}_{\text{Ref}} - \mathbf{u}_{\text{SPINN}}\|_2$. This visualization highlights RTNN's enhanced accuracy in capturing the complex magneto-fluid dynamic structures compared to the SPINN baseline at the specified time.

| Method | rL$_2$ Error (Velocity) | rL$_2$ Error (B) | Wall Time (s) |
|---|---|---|---|
| **RTNN** | **2.34e-02** | **1.04e-01** | 2340.41 |
| Curl-SPINN | 1.61e-01 | 1.55e-01 | 1191.67 |
| SPINN | 1.82e-01 | 2.93e-01 | **414.73** |

*Table 3.* Comparison of methods for the MHD experiment, reporting the median relative $L_2$ error for velocity and magnetic fields (B in the table), and median wall time across five independent training runs with different random seeds.

**Experimental setup** We train and validate our method on a three-dimensional *periodic* incompressible MHD flow in $[0, L]^2$ with Reynolds numbers $\text{Re} = 1000$ and $\text{ReM} = 1000$. The simulation covers the time interval $t \in [0, 0.5]$ for training and tests at $t = 0.55$. Training data is generated using a spectral solver, with initial velocity $\mathbf{u}_0$ and magnetic field $\mathbf{B}_0$ sampled from Gaussian random fields. Periodic boundary conditions are strictly enforced in all cases using the approach described in (Dong & Ni, 2021). We employ a *Separable Physics-Informed Neural Network* (SPINN) (Cho et al., 2023) comprising **5 hidden layers** and **500 neurons per layer** to handle the structured 3D grid, discretized into $\mathbf{101 \times 128 \times 128}$ points. The model is trained using **50,000 iterations** of the L-BFGS optimizer. Validation at $t = 0.55$ utilizes spectral solver data to evaluate performance. We compare three approaches: **(i)** our RTNN-based method, **(ii)** a SPINN baseline with penalized residuals, and **(iii)** a Curl-SPINN parametrizing velocity and magnetic fields as the curl of a scalar potential. Additional details and performance plots are provided in Appendix B.5.

**Results and Discussion** Table 3.4 reports the relative $L^2$ errors for both velocity and magnetic fields and Figure 7 shows the error evolution through training accross seeds. A qualitative comparison of the predicted velocity fields is presented in Figure 3, which visually underscores the enhanced accuracy of RTNN in capturing the complex magneto-fluid dynamic structures compared to the Curl-SPINN baseline. Our RTNN significantly improves velocity accuracy compared to the baselines and also enhances the accuracy of magnetic field predictions. Additionally, we demonstrated that RTNN can be incorporated into coupled systems for more complex problems.

## 4. Discussion and Conclusion

We introduced *Riemann Tensor Neural Networks* (RTNNs), a new class of neural architectures tailored for encoding divergence-free symmetric tensors (DFSTs). By construction, RTNNs exactly satisfy the DSFT conditions that encodes conservation of mass and momentum. Our theoretical results confirm that RTNNs are universal approximators of DFSTs, and our numerical benchmarks illustrate that they consistently improve accuracy compared to baselines such as standard PINNs and methods enforcing only part of the conservation (e.g., mass alone).

**Limitations and Future Work** While our experiments focus on fluid dynamics applications using MLPs, RTNNs have the potential to extend to diverse systems like Euler–Fourier, relativistic Euler, Boltzmann, to name but a few. Future work includes enhancing algorithm performance, extending RTNNs to operator learning frameworks, and reducing computational costs by developing architectures with more efficient differential operators. Additionally, exploring function space optimization techniques (Jnini et al., 2024b; Jnini & Vella, 2025) could further improve the accuracy of RTNNs.

## Acknowledgments

A.J. acknowledges support from a fellowship provided by Leonardo S.p.A. This work was partially funded under the NRRP, Mission 4 Component 2 Investment 1.4, by the European Union – NextGenerationEU (proj. nr. CN 00000013).

## Impact Statement

This paper presents work whose goal is to advance the field of Machine Learning. There are many potential societal consequences of our work, none which we feel must be specifically highlighted here.

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

# A. Proofs

## A.1. Proof of Theorem 2.1

In this section, we prove the main result on the representation of divergence-free symmetric tensors (Theorem A.1), which states that $\{S_{ab}\}$ (symmetric and divergence-free) exactly coincides with the image of the map $K_{acbd} \mapsto \nabla^c \nabla^d K_{acbd}$, where $K_{acbd}$ satisfies Riemann-like symmetries.

We first recall the classification of Riemann-like $(0,4)$-tensors (Lemma A.2), and then prove the surjectivity of the map $\Phi \colon K_{acbd} \mapsto \nabla^c \nabla^d K_{acbd}$ (Lemma A.4). Combining these two ingredients completes the proof of Theorem A.1.

### A.1.1. MAIN THEOREM: REPRESENTATION OF DIVERGENCE-FREE SYMMETRIC TENSORS

**Theorem A.1** (Representation of Divergence-Free Symmetric Tensors on a Flat Manifold). *Let $V$ be an $n$-dimensional real vector space with a fixed basis $\{e_a\}_{a=1}^n$, and let $\{e_a^*\}_{a=1}^n$ denote the corresponding dual basis of $V^*$. Let $\Lambda^2 V^*$ denote the space of 2-forms on $V$. Consider the space of all $(0,4)$-tensors $K_{abcd}$ defined on a flat manifold equipped with a Levi-Civita connection $\nabla$, satisfying the following symmetries:*

1. *Antisymmetry within index pairs:*
$$K_{(ab)cd} \; = \; 0, \quad K_{ab(cd)} \; = \; 0,$$

2. *Symmetry between pairs:*
$$K_{abcd} \; = \; K_{cdab}.$$

*Let $\{\omega_1, \ldots, \omega_m\}$ be a fixed basis of $\Lambda^2 V^*$, where $m = \frac{n(n-1)}{2}$. Define the tensors:*

$$T_{abcd}^{(i,j)} \; := \; \omega_i(e_a^* \wedge e_b^*) \, \omega_j(e_c^* \wedge e_d^*) \; + \; \omega_j(e_a^* \wedge e_b^*) \, \omega_i(e_c^* \wedge e_d^*).$$

*Then:*

1. *The space of* divergence-free, *symmetric $(0,2)$-tensors $S_{ab}$ on the flat manifold is exactly the image of the map*

$$K_{acbd} \; \longmapsto \; S_{ab} \; = \; \nabla^c \nabla^d K_{acbd}.$$

2. *Moreover, any such $S_{ab}$ can be expressed as*

$$S_{ab} \; = \; \sum_{1 \leq i \leq j \leq m} T_{acbd}^{(i,j)} \, \nabla^c \nabla^d c_{ij},$$

*where $c_{ij}$ are smooth scalar functions.*

**Proof Outline.** By Lemma A.2, every $K_{acbd}$ with the above symmetries can be expanded in the basis $\{T^{(i,j)}\}$. Hence $\nabla^c \nabla^d K_{acbd}$ can be written in the claimed form. We also check that $\nabla^c \nabla^d K_{acbd}$ is symmetric and divergence-free using flatness (commuting covariant derivatives) plus the antisymmetries. Finally, Lemma A.4 shows that *any* symmetric, divergence-free $S_{ab}$ arises from some $K_{acbd}$, establishing that the image of the map is the space of such $S_{ab}$.

### A.1.2. CLASSIFICATION OF RIEMANN-LIKE TENSORS

**Lemma A.2** (Classification of Riemann-like $(0,4)$-Tensors). *Let $V$ be an $n$-dimensional real vector space, and let $\Lambda^2 V^*$ be the space of 2-forms on $V$. Denote*

$$m \; = \; \dim(\Lambda^2 V^*) \; = \; \frac{n(n-1)}{2}.$$

*Consider the vector space*

$$\left\{ T_{abcd} \; \mid \; T_{abcd} = -T_{bacd}, \quad T_{abcd} = -T_{abdc}, \quad T_{abcd} = T_{cdab} \right\}.$$

*There is a canonical vector-space isomorphism between this space and* $\mathrm{Sym}^2(\Lambda^2 V^*)$. *In particular, its dimension is*

$$\frac{m(m+1)}{2} = \frac{\frac{n(n-1)}{2}\left(\frac{n(n-1)}{2}+1\right)}{2}.$$

*Moreover, if* $\{\omega_1, \ldots, \omega_m\}$ *is a basis for* $\Lambda^2 V^*$, *then a corresponding basis in the space of such* $(0,4)$-*tensors is given by*

$$T^{(i,j)}_{abcd} = \left(\omega_i \otimes \omega_j + \omega_j \otimes \omega_i\right)\left(e_a^* \wedge e_b^*, \, e_c^* \wedge e_d^*\right), \quad 1 \le i \le j \le m.$$

*Proof of Lemma A.2.* **Step 1: From** $T$ **to a symmetric bilinear form.** Given $T_{abcd}$ with the stated symmetries, define for $\alpha, \beta \in \Lambda^2 V^*$:

$$\alpha = \alpha^{ab}\, e_a^* \wedge e_b^*, \quad \beta = \beta^{cd}\, e_c^* \wedge e_d^*,$$
$$\widetilde{T}(\alpha, \beta) = T_{abcd}\, \alpha^{ab}\, \beta^{cd}.$$

Since $\alpha^{ab}$ and $\beta^{cd}$ are antisymmetric in their respective index pairs, and $T_{abcd}$ is antisymmetric in $(a,b)$ and $(c,d)$, this is well-defined. The symmetry $T_{abcd} = T_{cdab}$ implies that $\widetilde{T}$ is symmetric as a bilinear form on $\Lambda^2 V^*$. Hence $\widetilde{T} \in \mathrm{Sym}^2(\Lambda^2 V^*)$.

**Step 2: From a symmetric bilinear form back to** $T$. Conversely, given $\widetilde{T} \in \mathrm{Sym}^2(\Lambda^2 V^*)$, define

$$T_{abcd} = \widetilde{T}\left(e_a^* \wedge e_b^*, \, e_c^* \wedge e_d^*\right).$$

A straightforward check shows that $T_{abcd}$ inherits antisymmetry in $(a,b)$, antisymmetry in $(c,d)$, and the pair-exchange symmetry $(a,b) \leftrightarrow (c,d)$.

**Step 3: Isomorphism and basis.** These two constructions are linear inverses of each other, yielding a vector-space isomorphism between our $(0,4)$-tensors and $\mathrm{Sym}^2(\Lambda^2 V^*)$. The dimension follows from standard linear algebra. If $\{\omega_1, \ldots, \omega_m\}$ is a basis of $\Lambda^2 V^*$, then $\{\omega_i \otimes \omega_j + \omega_j \otimes \omega_i : 1 \le i \le j \le m\}$ forms a basis for $\mathrm{Sym}^2(\Lambda^2 V^*)$. Mapping these to $(0,4)$-tensors via the above correspondence yields the stated basis $\{T^{(i,j)}\}$. □

A.1.3. SURJECTIVITY OF THE MAP $\Phi\colon K \mapsto \nabla^c \nabla^d K$

As discussed, the second key ingredient for Theorem A.1 is showing that *every* divergence-free, symmetric $(0,2)$-tensor $S_{ab}$ can be obtained from some Riemann-like $(0,4)$-tensor $K_{acbd}$. Equivalently, the map

$$\Phi\colon K_{acbd} \mapsto S_{ab} = \nabla^c \nabla^d K_{acbd}$$

is surjective.

**Lemma A.3** (Surjectivity of $\Phi$). *Let* $K_{acbd}$ *be any Riemann-like* $(0,4)$-*tensor (i.e. satisfying the symmetries of Lemma A.2). Define* $S_{ab} = \nabla^c \nabla^d K_{acbd}$. *Then* $S_{ab}$ *is automatically symmetric and divergence-free. Moreover,* $\Phi$ *is onto: for any given symmetric, divergence-free* $(0,2)$-*tensor* $S_{ab}$, *there exists a Riemann-like* $K_{acbd}$ *such that* $S_{ab} = \nabla^c \nabla^d K_{acbd}$.

*Proof.* **Part 1:** We first verify that if $S_{ab} = \nabla^c \nabla^d K_{acbd}$, then $S_{ab}$ is symmetric and divergence-free.

- *Symmetry*: Using $K_{acbd} = K_{bdac}$, we get

$$S_{ab} = \nabla^c \nabla^d\, K_{acbd} = \nabla^c \nabla^d\, K_{bdac} = S_{ba}.$$

- *Divergence-free*: In flat space, covariant derivatives commute, so

$$\nabla^a S_{ab} = \nabla^a \nabla^c \nabla^d\, K_{acbd} = \nabla^c \nabla^d \nabla^a\, K_{acbd}.$$

Because $K_{acbd}$ is antisymmetric in $(a,c)$, one shows that $\nabla^a K_{acbd} = 0$. Hence $\nabla^a S_{ab} = 0$. □

**Part 2 (Surjectivity)**: We now show that given any symmetric, divergence-free $S_{ab}$, we can solve $S_{ab} = \nabla^c \nabla^d K_{acbd}$ for some Riemann-like $K_{acbd}$.

## A.2. Surjectivity of the map

**Theorem A.4** (Surjectivity of the $\Phi$). *We will now prove that the following map $\Phi$ is surjective:*

$$\Phi : K \mapsto S \tag{22}$$

$$S_{ab} = \Phi(K_{acbd}) = \nabla^c \nabla^d K_{acbd} \tag{23}$$

*where $K_{acbd}$ is a Riemann-like $(0,4)$ tensor and $S_{ab}$ is a symmetric divergence-free $(0,2)$ tensor.*

*Proof.* This is equivalent to showing that for any given $S_{ab}$, there exists a $K_{acbd}$ such that $S_{ab} = \Phi(K_{acbd})$. As in the previous sections, we assume we are working on a flat manifold.

Let's define a Poisson's equation (Griffiths, 2023) for each index of the tensor $S_{ab}$, with appropriate boundary conditions as desired:

$$\Delta L_{ab} = S_{ab} \tag{24}$$

These Poisson's equations can be solved by the well-known method of Green's function (Griffiths, 2023).

By solving the previous equations, we obtain a tensor $L_{ab}$ with the following properties:

$$L_{ab} = L_{ba} \tag{25a}$$

$$\nabla^a L_{ab} = 0 \tag{25b}$$

$$S_{ab} = \Delta L_{ab} \tag{25c}$$

It's easy to show that:

- Eq. 25a follows from the fact that the Laplacian operator $\Delta$ in Eq. 24 is meant componentwise and $S_{ab}$ is symmetric.

- Eq. 25b follows from the fact that the derivatives commute on a flat manifold.

- Eq. 25c follows form definition 24.

We can now easily define a tensor $K_{acbd}$ with the desired properties from $L_{ab}$. Just let:

$$K_{acbd} = 2 \left( \delta_{a[b} L_{d]c} - \delta_{c[b} L_{d]a} \right) \tag{26}$$
$$= \delta_{ab} L_{dc} - \delta_{ad} L_{bc} - \delta_{cb} L_{da} + \delta_{cd} L_{ba}$$

We now verify the Riemann-like properties of $K$:

$$K_{(ac)bd} = 4 \left( \delta_{a[b} L_{d]c} - \delta_{c[b} L_{d]a} + \delta_{c[b} L_{d]a} - \delta_{a[b} L_{d]c} \right) = 0 \tag{27}$$

$$K_{ac(bd)} = 4 \left( \delta_{a[b} L_{d]c} - \delta_{c[b} L_{d]a} + \delta_{a[d} L_{b]c} - \delta_{c[d} L_{b]a} \right) = 0 \tag{28}$$

$$K_{acbd} = 2 \left( \delta_{a[b} L_{d]c} - \delta_{c[b} L_{d]a} \right) \tag{29}$$
$$= \delta_{ab} L_{dc} - \delta_{ad} L_{bc} - \delta_{cb} L_{da} + \delta_{cd} L_{ba}$$
$$= \delta_{ba} L_{cd} - \delta_{da} L_{cb} - \delta_{bc} L_{ad} + \delta_{dc} L_{ab}$$
$$= \delta_{ba} L_{cd} - \delta_{bc} L_{ad} - \delta_{da} L_{cb} + \delta_{dc} L_{ab}$$
$$= 2 \left( \delta_{b[a} L_{c]d} - \delta_{d[a} L_{d]c} \right)$$
$$= K_{bdac}$$

$$\nabla^c \nabla^d K_{acbd} = 2 \nabla^c \nabla^d \left( \delta_{a[b} L_{d]c} - \delta_{c[b} L_{d]a} \right) \tag{30}$$
$$= \nabla^c \nabla^d \left( \delta_{ab} L_{dc} \right) - \nabla^c \nabla^d \left( \delta_{ad} L_{bc} \right) - \nabla^c \nabla^d \left( \delta_{cb} L_{da} \right) + \nabla^c \nabla^d \left( \delta_{cd} L_{ba} \right)$$
$$= \delta_{ab} \nabla^c \nabla^d L_{dc} - \delta_{ad} \nabla^c \nabla^d L_{bc} - \delta_{cb} \nabla^c \nabla^d L_{da} + \delta_{cd} \nabla^c \nabla^d L_{ba}$$
$$= 0 - 0 - 0 + \nabla^c \nabla_c L_{ba}$$
$$= \Delta L_{ab}$$
$$= S_{ab}$$

So, since we can build an inverse for every $S_{ab}$, the map $\Phi : K_{acbd} \mapsto S_{ab}$ is surjective.

$\square$

### A.2.1. COMBINING THE LEMMAS TO PROVE THEOREM A.1

*Proof of Theorem A.1.* **(1) Symmetry and divergence-free for** $\nabla^c\nabla^d K_{acbd}$**.** By Lemma A.4 (Part 1), the tensor $\nabla^c\nabla^d K_{acbd}$ is always symmetric and divergence-free if $K_{acbd}$ is Riemann-like.

**(2) Surjectivity: Every symmetric, divergence-free $S_{ab}$ arises from some $K_{acbd}$.** By Lemma A.4 (Part 2), the map $\Phi\colon K \mapsto \nabla^c\nabla^d K$ is onto the space of such $S_{ab}$.

**(3) Basis and explicit representation.** From Lemma A.2, any Riemann-like $K_{acbd}$ can be expanded in the basis $\{T^{(i,j)}\}$. Consequently,

$$\nabla^c\nabla^d K_{acbd} \;=\; \sum_{1 \le i \le j \le m} T^{(i,j)}_{acbd} \, \nabla^c\nabla^d c_{ij}$$

for some scalar functions $\{c_{ij}\}$. This shows that *every* $S_{ab}$ can indeed be written in the claimed form.

Hence all parts of the statement hold, and the proof is complete. $\square$

### A.3. Proof of theorem A.5

**Theorem A.5** (Universal Approximation for RTNN). *Let $\Omega \subset \mathbb{R}^3$ be a compact domain, and let $S \in C(\Omega; \mathbb{R}^{3 \times 3})$ be any divergence-free symmetric tensor field on $\Omega$. For any $\epsilon > 0$, there exists a Riemann Tensor Neural Network $S_\theta$ with a fixed narrow width and arbitrary depth, such that*

$$\sup_{x \in \Omega} \|S(x) - S_\theta(x)\| < \epsilon, \tag{31}$$

*where $\|\cdot\|$ denotes the Frobenius norm.*

*Proof.* To prove Theorem A.5, we proceed in two main steps:

1. **Surjectivity of the Map from $K$ to $S$:** By Theorem A.4, the map

$$\Phi : K \mapsto S, \tag{32}$$

   defined by

$$S_{ab} = \nabla^c\nabla^d K_{acbd}, \tag{33}$$

   is surjective. This implies that for any divergence-free symmetric tensor field $S \in C(\Omega; \mathbb{R}^{m \times m})$, there exists a tensor $K$ with Riemann-like symmetries such that

$$S = \Phi(K). \tag{34}$$

2. **Approximation of Tensor $K$ Using Neural Networks:** Since $K$ is a linear combination of scalar functions, specifically

$$K_{acbd} = \sum_{i,j} c_{ij}(x) T^{(i,j)}_{acbd}, \tag{35}$$

   where $T^{(i,j)}_{acbd}$ are fixed basis tensors and $c_{ij}(x)$ are scalar coefficient functions, we can approximate each $c_{ij}(x)$ using deep narrow neural networks.

   By the Universal Approximation Theorem for deep narrow neural networks (Kidger & Lyons, 2020), for each scalar function $c_{ij}(x)$ and for any $\epsilon > 0$, there exists a neural network $\text{NN}_{\theta_{ij}}$ such that

$$\sup_{x \in \Omega} \left| c_{ij}(x) - c^{\text{NN}}_{ij}(x; \theta_{ij}) \right| < \frac{\epsilon}{C}, \tag{36}$$

where $C$ is a constant dependent on the norms of the basis tensors $T^{(i,j)}$ and the chosen tensor norm $\|\cdot\|$.

Construct the approximated tensor $K_\theta$ as

$$K_\theta(x) = \sum_{i,j} c_{ij}^{\text{NN}}(x;\theta_{ij})T_{acbd}^{(i,j)}. \tag{37}$$

Applying the map $\Phi$ to $K_\theta$, we obtain the approximated tensor $S_\theta$:

$$S_\theta(x) = \Phi(K_\theta)(x) = \nabla^c\nabla^d K_\theta(x). \tag{38}$$

**Error Propagation:**

The approximation error in each $c_{ij}(x)$ propagates linearly to $S_\theta(x)$. Specifically, we have

$$
\begin{aligned}
\|S(x) - S_\theta(x)\| = & \tag{39}\\
= & \|\Phi(K)(x) - \Phi(K_\theta)(x)\|\\
= & \left\|\nabla^c\nabla^d(K - K_\theta)(x)\right\|\\
= & \left\|\nabla^c\nabla^d\left(\sum_{i,j}(c_{ij}(x) - c_{ij}^{\text{NN}}(x;\theta_{ij}))T_{acbd}^{(i,j)}\right)\right\|\\
\leq & \sum_{i,j}\|T^{(i,j)}\|\sup_{x\in\Omega}\left|\nabla^c\nabla^d\left(c_{ij}(x) - c_{ij}^{\text{NN}}(x;\theta_{ij})\right)\right|\\
\leq & \sum_{i,j}\|T^{(i,j)}\|\sup_{x\in\Omega}\left|c_{ij}(x) - c_{ij}^{\text{NN}}(x;\theta_{ij})\right|\cdot C_{ij}\\
< & \sum_{i,j}\|T^{(i,j)}\|\cdot\frac{\epsilon}{C}\\
= & \epsilon,
\end{aligned}
$$

where $C_{ij}$ accounts for the bounds on the second-order partial derivatives $\nabla^c\nabla^d c_{ij}(x)$ over the compact domain $\Omega$, and the final equality holds by the choice of $C$.

Therefore, by appropriately choosing the neural networks $\text{NN}_{\theta_{ij}}$ to approximate each scalar coefficient $c_{ij}(x)$ within $\epsilon/C$, we ensure that the approximated tensor $S_\theta(x)$ satisfies

$$\sup_{x\in\Omega}\|S(x) - S_\theta(x)\| < \epsilon. \tag{40}$$

$\square$

## A.4. Equivalence of Explicit Momentum Residual and Isotropic Loss Term

Consider the compressible Euler momentum equation (neglecting viscous terms):

$$\partial_t(\rho u) + \nabla\cdot(\rho u\otimes u + pI) = 0,$$

where $\rho$ is the density, $u$ is the velocity field, and $p$ is the pressure.

In our RTNN parametrization, the network outputs a flux tensor $S$ defined as:

$$S = \begin{pmatrix}\rho & (\rho u)^T\\ \rho u & \rho u\otimes u + \sigma\end{pmatrix},$$

with $\rho = S_{0,0}$ and $\rho u = S_{1,0}$. The stress tensor is then defined by:

$$\sigma = S_{1:2,1:2} - \rho\,u\otimes u.$$

We impose a zero-deviatoric constraint on $\sigma$, meaning its deviatoric part vanishes:

$$\sigma_{\mathrm{dev}} \triangleq \sigma - \frac{1}{n}\,\mathrm{tr}(\sigma)\,I = 0.$$

This forces:

$$\sigma = \frac{1}{n}\,\mathrm{tr}(\sigma)\,I \triangleq pI,$$

With the pressure defined as:

$$p = \frac{1}{n}\,\mathrm{tr}(\sigma).$$

Substituting $\sigma = pI$ back into the flux tensor, we get:

$$S = \begin{pmatrix} \rho & (\rho u)^T \\ \rho u & \rho u \otimes u + pI \end{pmatrix}.$$

Our architecture enforces that $S$ is divergence-free:

$$\nabla \cdot S = \begin{pmatrix} \partial_t \rho + \nabla \cdot (\rho u) \\ \partial_t(\rho u) + \nabla \cdot (\rho u \otimes u + pI) \end{pmatrix} = 0.$$

The first row yields the continuity equation, which vanishes:

$$\partial_t \rho + \nabla \cdot (\rho u) = 0.$$

The second row yields:

$$\partial_t(\rho u) + \nabla \cdot (\rho u \otimes u + pI) = 0,$$

which, when expanded, yield the following equations:

$$\partial_t \rho + \nabla \cdot (\rho u, \rho v) = 0 \tag{41}$$

$$\partial_t(\rho u) + \nabla \cdot (\rho u^2, \rho uv) = -\partial_x p \tag{42}$$

$$\partial_t(\rho v) + \nabla \cdot (\rho uv, \rho v^2) = -\partial_y p \tag{43}$$

This is exactly the system in (11)–(13).

In other words, the condition on the stress tensor is numerically equivalent to having the momentum residual vanish. A similar reasoning can be applied when adding viscous terms (or magnetic-terms), for instance.

Furthermore, to prove our mathematical derivation numerically, we conducted an experiment in which the velocity and pressure fields extracted from a single RTNN were trained using the classical momentum equation rather than the simplified condition on the stress tensor. In this setting, the only difference among the experiments is the underlying architecture. Although this process is more computationally expensive, owing to the necessity of computing a fourth-order derivative, the RTNN-trained fields still demonstrate similarly superior accuracy compared to those obtained with PINN and NCL. We summarize the obtained results below for the two unsteady Navier–Stokes problems: flow around a cylinder and the Beltrami flow (see Table 4).

PINN and NCL were both trained using the full system (11)–(14), which includes the conservation of mass, momentum, and energy equations. For NCL, the mass conservation equation is satisfied by construction and cancels up to machine precision; similarly, for RTNN, conservation of mass is similarly enforced structurally and cancels out. The energy equation (14) is also included in the loss, similarly to the other methods.

The only apparent difference is that RTNN replaces the explicit momentum residual with an isotropic loss term. However, this isotropic loss term naturally emerges when substituting the RTNN-predicted fields into the flux-form momentum equation. Imposing the zero-deviatoric condition on the stress tensor is numerically equivalent to enforcing the momentum equation directly in its expanded form.

| Method | Beltrami | | Cylinder | |
|---|---|---|---|---|
| | **L2 Median** | **L2 IQR** | **L2 Median** | **L2 IQR** |
| RTNN (extended momentum equation) | 4.55e-04 | 8.1e-05 | 5.13e-03 | 4.2e-04 |
| RTNN (stress tensor) | 4.28e-04 | 5.9e-05 | 5.70e-03 | 4.3e-04 |
| NCL | 1.74e-03 | 8.5e-05 | 2.54e-02 | 1.9e-03 |
| PINN | 1.41e-03 | 3.1e-04 | 2.99e-02 | 7.9e-04 |

*Table 4.* Comparison of methods across Beltrami and Cylinder tasks. Reported: median and IQR of relative L2 error.

## B. Additional resources for Experiments

### B.1. Euler Equation

The simulation of the 2D isentropic Euler vortex is based on analytical solutions, ensuring precise modeling of vortex dynamics. Below, we provide detailed initial and boundary conditions, as well as a summary of the experimental setup parameters and hyperparameters.

*Table 5.* Setup and Hyperparameters for the Euler Vortex Experiment

| Parameter | Value |
|---|---|
| Optimizers | BFGS |
| Architecture | 4-layer Multilayer Perceptron, width 50 |
| Activation Function | Tanh |
| Domain | $\Omega = [0, L_x] \times [0, L_y]$ |
| Time Interval | $[0, T]$ |
| Collocation Points (Interior) | 1,000 |
| Collocation Points (Boundary) | 200 |
| Validation Points | 10,000 |
| Evaluation Metric | Relative $L^2$ Error |

**Initial Conditions:** The initial density, velocity components, and temperature distributions are defined based on the analytical solution of the Euler vortex:

$$\rho(x, y, 0) = \rho_\infty \left( \frac{T(x, y)}{T_\infty} \right)^{1/(\gamma-1)}, \tag{44}$$

$$u(x, y, 0) = u_\infty - \frac{\beta}{2\pi}(y - y_c) \exp\left[1 - r^2(x, y)\right], \tag{45}$$

$$v(x, y, 0) = v_\infty + \frac{\beta}{2\pi}(x - x_c) \exp\left[1 - r^2(x, y)\right], \tag{46}$$

$$T(x, y) = T_\infty - \frac{(\gamma - 1)\beta^2}{8\pi^2} \exp\left[2(1 - r^2(x, y))\right], \tag{47}$$

where $r^2(x, y) = (x - x_c)^2 + (y - y_c)^2$ represents the radial distance from the vortex center $(x_c, y_c)$.

**Boundary Conditions:** On the boundaries of the spatial domain $\partial\Omega$, we impose fixed values for density, velocity, and pressure:

$$\rho|_{\partial\Omega} = \rho_\infty, \quad (u, v)|_{\partial\Omega} = (u_\infty, v_\infty), \quad p|_{\partial\Omega} = \kappa\rho_\infty^\gamma.$$

## B.2. Beltrami Flow

Table 6. Setup and Hyperparameters for the Beltrami Flow Experiment

| Parameter | Value |
| --- | --- |
| Optimizers | BFGS |
| Architecture | 4-layer Multilayer Perceptron, width 50 |
| Activation Function | Tanh |
| Domain | $\Omega = [-1, 1] \times [-1, 1] \times [-1, 1]$ |
| Time Interval | $[0, 1]$ |
| Collocation Points (Interior) | 10,000 |
| Collocation Points (Boundary) | 961 per face |
| Validation Points | 10,000 |
| Evaluation Metric | Relative $L^2$ Error |

We consider the unsteady three-dimensional Beltrami flow originally described by Ethier & Steinman (1994) at $\mathrm{Re} = 1$. The spatial domain is $\Omega = [-1, 1]^3$ and time ranges over $[0, 1]$. The exact solutions are:

$$u(x, y, z, t) = -e^x \sin(y + z) + e^z \cos(x + y) \, e^{-t},$$
$$v(x, y, z, t) = -e^y \sin(z + x) + e^x \cos(y + z) \, e^{-t},$$
$$w(x, y, z, t) = -e^z \sin(x + y) + e^y \cos(z + x) \, e^{-t},$$

and

$$p(x, y, z, t) = -\tfrac{1}{2} \big[ e^{2x} + e^{2y} + e^{2z} + 2 \sin(x + y) \, \cos(z + x) \, e^{y+z} \\ + 2 \sin(y + z) \, \cos(x + y) \, e^{z+x} + 2 \sin(z + x) \, \cos(y + z) \, e^{x+y} \big] e^{-2t}.$$

These expressions satisfy the incompressible Navier–Stokes equations exactly, making the Beltrami flow an ideal test for verifying numerical methods. We use 10,000 interior collocation points to enforce the PDE residual and 961 boundary/initial points per face for constraints; an additional 10,000 points in the interior serve as a validation set. We report the training dynamics in Figure B.2.

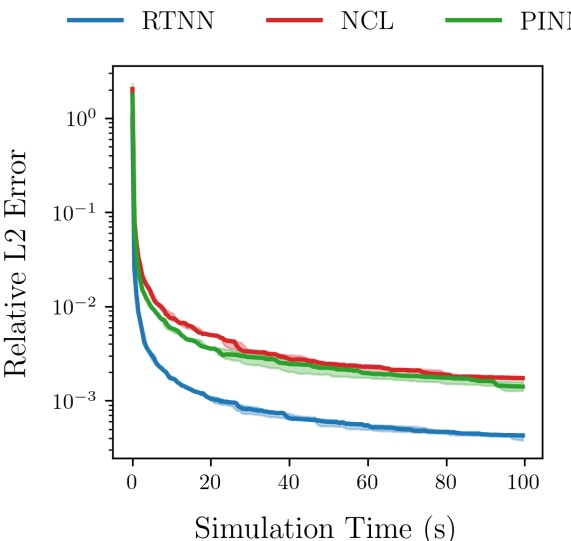

*Figure 4.* Comparison of training relative $L^2$ error over simulation time for the 3d Beltrami experiment across all methods. The solid lines represent the median relative $L^2$ error, while the shaded regions indicate the interquartile range (IQR). RTNN achieves the lowest L2 error, followed by PINN and NCL. The x-axis represents the simulation time (in seconds), while the y-axis shows the relative $L^2$ error on a logarithmic scale.

## B.3. Cylinder Wake

*Table 7.* Setup and Hyperparameters for the Cylinder Wake Experiment

| Parameter | Value |
|---|---|
| Optimizers | BFGS |
| Architecture | 4-layer Multilayer Perceptron, width 50 |
| Activation Function | Tanh |
| Domain | $\Omega = [1, 8] \times [-2, 2]$ |
| Time Interval | $[0, 7], \Delta t = 0.1$ |
| Interior Collocation Points | 40,000 |
| Boundary/Initial Points | 5,000 |
| Evaluation Metric | Relative $L^2$ Error |

For the two-dimensional cylinder wake at $\mathrm{Re} = 100$, we consider a circular cylinder centered at $(0, 0)$. The downstream domain is $\Omega = [1, 8] \times [-2, 2]$ in space, with the simulation time interval $[0, 7]$ discretized in increments of $\Delta t = 0.1$. We train a 4-layer MLP (width 50, Tanh activations) on 40,000 interior collocation points and 5,000 boundary/initial points, *without* using labeled data. We validate the solution against direct numerical simulation (DNS) from Raissi et al. (2019a). We report the training dynamics in Figure 5

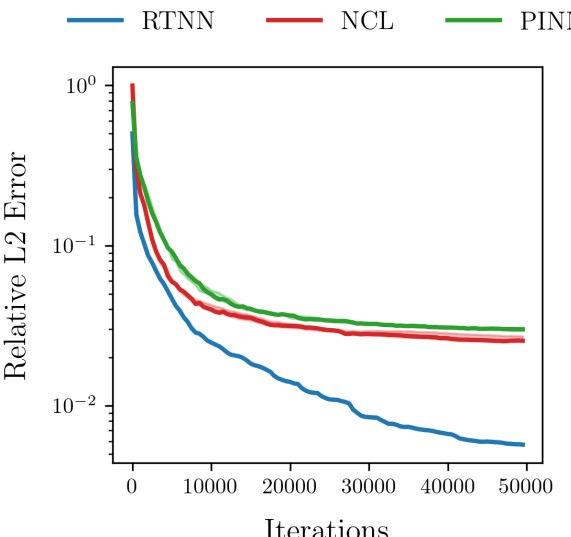

*Figure 5.* Comparison of training relative $L^2$ error over simulation time for the Cylinder wake experiment across all methods. The solid lines represent the median relative $L^2$ error, while the shaded regions indicate the interquartile range (IQR). RTNN achieves the lowest L2 error, followed by NCL and pinn . The x-axis represents the number of iterations, while the y-axis shows the relative $L^2$ error on a logarithmic scale.

### B.4. NACA Airfoil

*Table 8.* Setup and Hyperparameters for the NACA Airfoil Experiment

| Parameter | Value |
| --- | --- |
| Optimizers | BFGS |
| Architecture | 4-layer Multilayer Perceptron, width 50 |
| Activation Function | Tanh |
| Evaluation Metric | Relative $L^2$ Error |

We investigate the steady laminar flow at $\mathrm{Re} = 1000$ around a NACA 0012 airfoil. This problem is recast as a steady Navier–Stokes system by treating time as a dummy dimension. An MLP with 4 hidden layers (width 50, Tanh activations) is trained *with* labeled data from our in-house RANS solver with 2'000 data points and 40'000 collocation points for PDE residuals. We use 12'000 data points to validate the recovered fields. We report the training dynamics in Figure 6

### B.5. MHD

We train and validate our method on a three-dimensional *periodic* incompressible Magnetohydrodynamics (MHD) flow within the domain $[0, L]^3$, with Reynolds numbers $\mathrm{Re} = 1000$ and $\mathrm{ReM} = 1000$. The simulation covers the time interval $t \in [0, 0.5]$ for training and tests at $t = 0.55$. Training data is generated using a spectral solver, ensuring high accuracy, with initial velocity $\mathbf{u}_0$ and magnetic field $\mathbf{B}_0$ sampled from Gaussian random fields. Periodic boundary conditions are strictly enforced to maintain physical consistency across the domain. We employ a *Separable Physics-Informed Neural Network* (SPINN) (Cho et al., 2023) with **5 hidden layers** and **500 neurons per layer** to handle the structured 3D grid, discretized into **101 × 128 × 128** points along each respective dimension. The model is trained using **50,000 iterations** of the L-BFGS optimizer to effectively minimize the loss function. Validation at $t = 0.55$ utilizes spectral solver data to assess performance and ensure accurate replication of MHD flow characteristics. We compare three approaches: **(i)** our RTNN-based method,

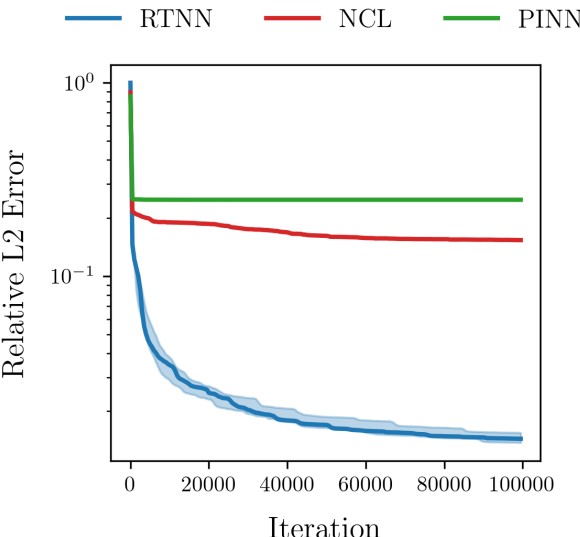

*Figure 6.* Comparison of training relative $L^2$ error over simulation time for the NACA Airfoil experiment across all methods. The solid lines represent the median relative $L^2$ error, while the shaded regions indicate the interquartile range (IQR). RTNN achieves the lowest L2 error, followed by PINN and NCL. The x-axis represents the simulation time (in seconds), while the y-axis shows the relative $L^2$ error on a logarithmic scale.

**(ii)** a SPINN baseline with penalized residuals, and **(iii)** a Curl-SPINN parametrizing velocity and magnetic fields as the curl of a scalar potential. Additional details and performance plots are provided below.

*Table 9.* Setup and Hyperparameters for the MHD Experiment

| Parameter | Value |
|---|---|
| **Optimizer** | L-BFGS |
| **Architecture** | Separable Physics-Informed Neural Network (SPINN) |
| **Hidden Layers** | 5 |
| **Neurons per Layer** | 500 |
| **Activation Function** | Tanh |
| **Domain** | $[0, L]^3$ |
| **Time Interval** | Training: $t \in [0, 0.5]$, Testing: $t = 0.55$ |
| **Grid Discretization** | $101 \times 128 \times 128$ points |
| **Reynolds Number** | $\mathrm{Re} = 1000$ |
| **Magnetic Reynolds Number** | $\mathrm{ReM} = 1000$ |
| **Training Iterations** | 50,000 |
| **Validation Data** | Spectral solver data at $t = 0.55$ |
| **Evaluation Metric** | Relative $L^2$ Error |

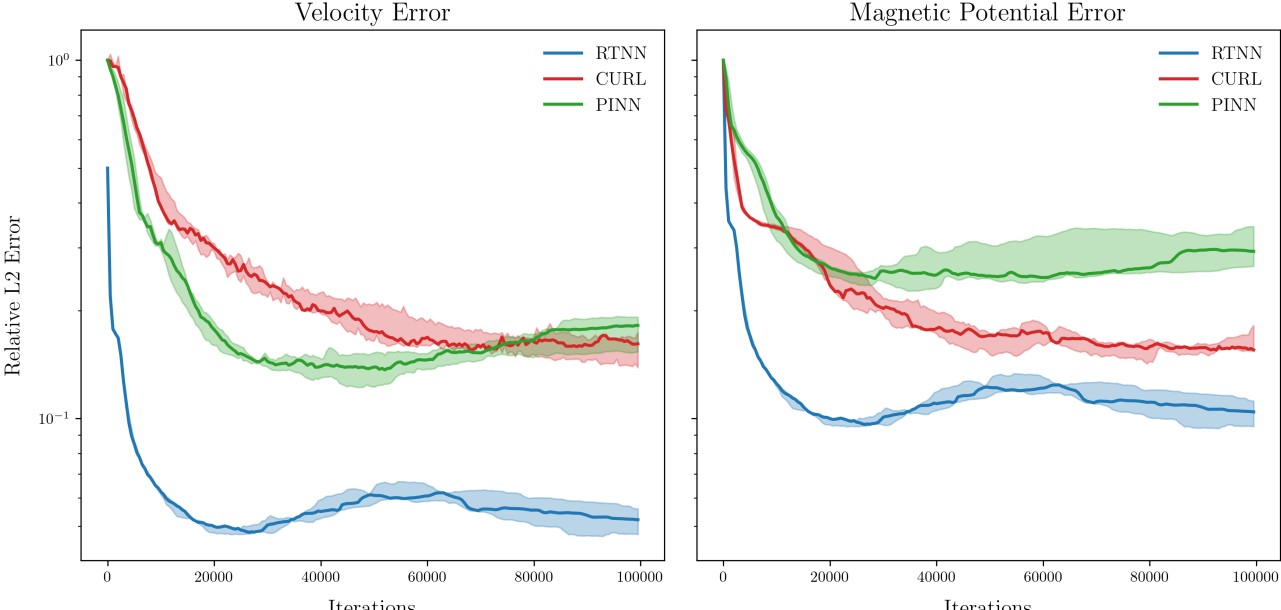

*Figure 7.* Comparison of training relative $L^2$ error over iterations for the MHD experiment across all methods. The left subplot represents the velocity relative $L^2$ error, while the right subplot represents the magnetic potential relative $L^2$ error. The solid lines indicate the median relative $L^2$ error, and the shaded regions depict the interquartile range (IQR). RTNN achieves the lowest $L^2$ error in both velocity and magnetic potential, followed by PINN and NCL. The x-axis represents the iterations, while the y-axis shows the relative $L^2$ error on a logarithmic scale.

