# OpenReview forum: "Riemann Tensor Neural Networks: Learning Conservative Systems with Physics-Constrained Networks"
_ICML.cc/2025/Conference — ICML 2025 poster_

### Official Review · Reviewer_1n14 · 2025-03-11

**Overall Recommendation:** 3

**Summary:**

This paper is within the setting of learning physical systems from PDE equations. The authors propose a neural network architecture that outputs a divergence-free symmetric tensor (DSFT). By adding this constraint, the neural network is guaranteed to enforce conservation of mass and momentum. In multiple experiments, the authors describe how to recover the solution from the DSFT output, after training the model on a loss function comprising PINNs loss, energy loss, and supervised L2 loss at some points. The authors compare their method to PINNs and Neural Conservation Laws (NCL), the latter of which is a similar method that only enforces conservation of mass. They also use their method to augment and compare to separable physics informed neural networks S-PINNs.

# Post rebuttal response
I raised my score from 2 to 3 after the authors' rebuttal (see my comment below).

**Claims And Evidence:**

The method is presented in a straightforward way and makes sense that it would be an improvement over unconstrained PINNs and NCLs which only conserve mass. The experiments appear sensible and show a marked improvement over the selected baselines.

**Essential References Not Discussed:**

I am not aware.

**Experimental Designs Or Analyses:**

I read through the design and setup of the experimental designs. I have a concern with how the competing methods are constructed. For example, in section 3.2, the loss function used to train the method is defined in terms of isotropic loss and energy conservation, rather than the PDE system described in (11)-(13). It seems that a different loss function is used for PINNs and NCL (line 247, column 2). Thus, it is unclear whether the performance increase is due to a different loss function or due to the novel architecture (the author's contribution).

**Methods And Evaluation Criteria:**

Yes, the evaluation criteria make sense. They compare to other common problems and show an order of magnitude improvement in each case. My expertise is not in PDEs, so I cannot attest to whether these problems are meaningful to that domain; this would be something I would defer to other reviewers on.

**Other Comments Or Suggestions:**

N/a

**Other Strengths And Weaknesses:**

I think the idea to build constraints into the architecture makes sense and is a good approach. If my concern about the experimental section is addressed, I would consider raising my score.

**Questions For Authors:**

Please see my comments under "Methods and Evaluation Criteria". While the results of your method appear strong, for the comparison to be fair the only difference between each comparator should be the architecture (i.e., PINNS=unconstrained, NCL=mass only, DSFT=mass and momentum). Please clarify whether this is the case in your experiments. If not, I request that you provide the results showing differences in architecture. Otherwise, it is unclear whether your method does better simply by using a specialized loss function for each problem (while NCL and PINNs do not).

**Relation To Broader Scientific Literature:**

This method offers a novel architecture to provide hard constraints on conservation of mass and momentum. It builds on methods like NCL by hard-coding physical invariants into the architecture.

**Theoretical Claims:**

I did not check the proofs of the theoretical claims.

---

> ### Author Rebuttal · Authors · 2025-03-31
>
> We thank the reviewer for their time reviewing the manuscript and their feedback that will contribute to improving the clarity of our work. Below we address the reviewer's concern:
>
> **Methods and Evaluation Criteria**:
>
>
> PINN and NCL were both trained using the full system (11)--(14), which includes the conservation of mass, momentum, and energy equations. For NCL , the mass conservation equation is satisfied by construction and cancels up to machine precision.
>
> For RTNN, the continuity equation similarly cancels out and the energy equation (14) is also included in the loss. The only apparent difference is that RTNN replaces the explicit momentum residual with an isotropic loss term. However, this isotropic loss term naturally emerges when substituting the RTNN-predicted fields into the flux-form momentum equation. Imposing the zero-deviatoric condition on the stress tensor is numerically equivalent to enforcing the momentum equation directly in its expanded form (12--13). This equivalence can be evidenced following the reasoning below:
>
> Consider the compressible Euler system in (11)--(14):
>
> In our RTNN parametrization, the network outputs a flux tensor \\(S\\) defined as:
>
> $$
> S = \\begin{pmatrix}
> \\rho & (\\rho u)^T \\\\
> \\rho u & \\rho u \\otimes u + \\sigma
> \\end{pmatrix},
> $$
>
> with \\(\\rho = S_{0,0}\\) and \\(\\rho u = S_{0,1:n}\\) . The stress tensor is then defined by:
>
> $$
> \\sigma = S_{1:n,1:n} \\; - \\; \\rho\\, u \\otimes u.
> $$
>
> We impose a zero-deviatoric constraint on \\(\\sigma\\), meaning its deviatoric part vanishes:
>
> $$
> \\sigma_{\\mathrm{dev}} \\;=\\; \\sigma \\; - \\; \\frac{1}{n}\\,\\operatorname{tr}(\\sigma)\\,I \\;\\triangleq  0.
> $$
>
> This forces:
>
> $$
> \\sigma \\triangleq \\frac{1}{n}\\,\\operatorname{tr}(\\sigma)\\,I \\quad \\triangleq \\quad pI,
> $$
>
> with the pressure defined as:
>
> $$
> p =\\ \\frac{1}{n}\\,\\operatorname{tr}(\\sigma).
> $$
>
> Substituting \\(\\sigma \\triangleq pI\\) back into the flux tensor, we obtain:
>
> $$
> S = \\begin{pmatrix}
> \\rho & (\\rho u)^T \\\\
> \\rho u & \\rho u \\otimes u + pI
> \\end{pmatrix}.
> $$
>
> Our architecture enforces that \\(S\\) is divergence-free:
>
> $$
> \\nabla \\cdot S = \\begin{pmatrix}
> \\partial_t \\rho + \\nabla \\cdot (\\rho u) \\\\[6pt]
> \\partial_t (\\rho u) + \\nabla \\cdot (\\rho u \\otimes u + pI)
> \\end{pmatrix} = 0.
> $$
>
> The first row yields the continuity equation:
>
> $$
> \\partial_t \\rho + \\nabla \\cdot (\\rho u) = 0.
> $$
>
> The second row yields:
>
> $$
> \\partial_t (\\rho u) + \\nabla \\cdot \\Bigl(\\rho u \\otimes u + pI\\Bigr) = 0,
> $$
>
> which when expanded yields the following equations:
>
> $$
> \\partial_t \\rho + \\nabla \\cdot (\\rho u, \\rho v) = 0 \\tag{11}
> $$
>
> $$
> \\partial_t (\\rho u) + \\nabla \\cdot (\\rho u^2, \\rho u\\,v) = -\\partial_x p \\tag{12}
> $$
>
> $$
> \\partial_t (\\rho v) + \\nabla \\cdot (\\rho u\\,v, \\rho v^2) = -\\partial_y p \\tag{13}
> $$
>
> This is exactly the system in (11--13).
>
> A similar reasoning can be applied when adding viscous terms (or magnetic terms), for instance.
>
> Furthermore, to address the reviewer's concern numerically, we conducted an experiment in which the velocity and pressure fields extracted from a single RTNN were trained using the expanded momentum (12)--(13) equation rather than the simplified condition on the stress tensor. In this setting, the only difference among the experiments is the architecture. Although this process is more computationally expensive --- owing to the necessity of computing a fourth-order derivative --- the RTNN-trained fields still demonstrate similarly superior accuracy compared to those obtained with PINN and NCL. We summarize the obtained results below for the two unsteady Navier--Stokes problems: flow around a cylinder and the Beltrami flow for five different initialization seeds:
>
> | Method                                | Beltrami rL2 (Median ± IQR)         | Cylinder rL2 (Median ± IQR)         |
> |---------------------------------------|-------------------------------------|-------------------------------------|
> | RTNN (expanded momentum equation)     | 4.55e-04 ± 8.1e-05                  | 5.13e-03 ± 4.2e-04                  |
> | RTNN (stress tensor condition)        | 4.28e-04 ± 5.9e-05                  | 5.70e-03 ± 4.3e-04                  |
> | NCL                                   | 1.73e-03 ± 8.5e-05                  | 2.54e-02 ± 1.9e-03                  |
> | PINN                                  | 1.41e-03 ± 3.1e-04                  | 2.99e-02 ± 7.9e-04                  |
>
> We acknowledge that the derivation of the condition on the stress tensors from the momentum equations could have been made clearer in the manuscript. We will appropriately make this link clear in the camera-ready version and provide the direct derivation in the appendix.
>
> We hope we have addressed the reviewer's main concerns and remain at their disposal should further clarifications be needed.

---

> > ### Comment · Reviewer_1n14 · 2025-04-01
> >
> > I appreciate the author's response. I appreciate the additional table showing the performance of RTNN using the same loss function as PINNs and NCL. I think this highlights the benefit of the architecture, because the error improvement is mostly the same.
> >
> > Also, your response implicitly shows the benefit of your RTNN reparameterization in avoiding the fourth-order derivative computation.
> >
> > I'm willing to raise my score, since my main concern from the review is addressed.

---

### Official Review · Reviewer_G2wL · 2025-03-14

**Overall Recommendation:** 4

**Summary:**

In this paper, a method for learning divergence-free symmetric tensors is proposed.  As neural network architectures that maintain conservation laws, a typical approach is learning divergence-free vector fields by using the exterior calculus. The proposed method is another approach based on a different geometric structure. As far as I know, the use of this structure is certainly new. In numerical experiments, it is shown that the proposed method achieves higher accuracy than a method for learning divergence-free vector fields. I think that the proposed approach is very interesting and powerful.

**Claims And Evidence:**

The method is clearly explained with its theoretical background.

**Essential References Not Discussed:**

Perhaps discussing the following paper, which is an operator learning method that is based on a similar approach to CLN, is not essential but helpful for understanding the contributions of this paper.

Ning Liu, Yiming Fan, Xianyi Zeng, Milan Klöwer, Lu Zhang, Yue Yu, Harnessing the Power of Neural Operators with Automatically Encoded Conservation Laws, https://arxiv.org/abs/2312.11176

**Experimental Designs Or Analyses:**

I believe that the experiments are designed appropriately; however, it seems strange that NCL performs worse than ordinary PINNs as seen in Table 1. In my understanding, similar to the proposed method, NCL is designed to satisfy conservation laws. However, the performance of one method (the proposed method) improves greatly while the other (NCL) declines, even though they are trying to maintain similar properties.

**Methods And Evaluation Criteria:**

I believe that the proposed method is evaluated in an appropriate manner.

**Other Comments Or Suggestions:**

None.

**Other Strengths And Weaknesses:**

Overall, I think that this is a very interesting paper.

**Questions For Authors:**

I want to confirm the main difference between NCL and the proposed method. In my understanding, it is difficult for NCL to preserve multiple conservatoin laws simultaniously, but the proposed method can preserve them. Is this understanding correct?

Also, why did NCL perform so poorly in the numerical experiment?

**Relation To Broader Scientific Literature:**

This paper proposes a new structure for neural networks that preserves conservation laws. In previous papers, the property d d = 0 of the exterior derivative has been mainly used. The approach in this paper is different from those in the existing papers and has a potential to improve neural networks for physics.

**Theoretical Claims:**

I checked the outline of the proofs, but I did not check them thoroughly.

---

> ### Author Rebuttal · Authors · 2025-03-31
>
> We would like to thank the reviewer for their time and feedback. Below we address some of the raised points:
>
> **Main Difference between NCL and RTNN:**
> NCL enforces only the continuity (mass-conservation) equation, whereas our approach reformulates the conservation laws in flux form, constraining each flux tensor to be both divergence-free and symmetric. This flux representation allows us to extract primitive fields—density, momenta, and stress tensors—that are geometrically coupled and, by construction, satisfy both mass and momentum conservation. In fact, when the RTNN-derived fields are substituted into the flux-form momentum equation, the momentum conservation learning problem is reduced to an optimization problem on the stress tensors.
>
> **On NCL’s Performance in the Compressible Experiment:**
> In our compressible experiment, NCL underperformed compared to other approaches. In this case, we have variable density and an additional energy equation, a scenario that differs from the incompressible cases where NCL performed well (both in our experiments and in the original NCL paper).
>
> Upon investigation we observed that, in the compressible setting, the NCL architecture struggled to simultaneously satisfy the PDE residual and the initial condition, likely due to imbalances in the loss terms. To verify this hypothesis, we transformed the ansatz to the following form:
>
> $$
> F(t, x) = \text{NCL}(t, x) - \text{NCL}(0, x) + u_0(x)
> $$
>
> Here, \\( u_0(x) \\) represents the initial condition. This formulation enforces both the continuity and initial conditions by design, resulting in significantly improved accuracy and confirming our conjecture, and also supports the argument that hard constraint satisfaction leads to simpler optimization problems.
>
> **Related References:**
> We reviewed the reference by Liu *et al.* [1] and find it both interesting and relevant to our work, we thank the reviewer for bringing it to our attention. We will add it to the related work section of the camera-ready version.
>
> We hope to have addressed the reviewer's question sufficiently and remain at their disposal throughout the review process.
>
>
> **References:**
>
> [1] Ning Liu, Yiming Fan, Xianyi Zeng, Milan Klöwer, Lu Zhang, Yue Yu, Harnessing the Power of Neural Operators with Automatically Encoded Conservation Laws, https://arxiv.org/abs/2312.11176

---

### Official Review · Reviewer_pPMm · 2025-03-14

**Overall Recommendation:** 5

**Summary:**

This paper concerns preserving certain structure properties when using neural networks to solve partial differential equations. The method uses an inductive bias by enforcing a certain form of the tensor field approximation, and the final form is called "Riemann Tensor Neural Network" (RTNN) involves some non-trainable final layer, plus an "differentiation layer" where the Hessian of the output from the previous layers is computed.

**Claims And Evidence:**

The authors show that RTNN is constructed to satisfy the divergence-free form of systems of conservation laws. This is shown by straightforward calculation.

In numerical experiments, the authors illustrate that RTNN can be used in a collocated PDE-related loss minimization framework similar to Physics-Informed Neural Networks (PINNs) framework to solve various PDEs.

**Essential References Not Discussed:**

This type of embedding of conservation laws appear to be new, as far as I know.

**Experimental Designs Or Analyses:**

The tested set of PDEs is significant that can be challenging for standard PINNs. However, it is not easy to see what kind of singular behaviors (sharp gradients) the solution has. Perhaps the authors can describe more in detail what kind of features are challenging to learn in these problems for existing methods like PINNs, and what contributes to the high errors for these methods (as opposed to RTNNs). There are some details in the current form of the manuscript, but it is hard to infer what the challenge is for the individual problems.

**Methods And Evaluation Criteria:**

The evaluation criteria is relative L2 error against reference solution, which is reasonable.

**Other Comments Or Suggestions:**

I suggest the authors show a few of the solution plots (rather than the error plots only) so the readers can infer the difficulty in PINNs solution

**Other Strengths And Weaknesses:**

The PDEs that appear in numerical examples are challenging PDEs for PINNs. The results show a clear advantage compared to standard PINNs.

The universal approximation theorem (Theorem 2.4) appears to be a very generic result, however, and does not shed a lot of light into why RTNNs have superior numerical performance.

**Questions For Authors:**

Is there any numerical issues for approximating the solution that contains what is essentially a "differentiation layer"? Repeated differentiations can result in loss of significant digits, and can be a source of instability.

**Relation To Broader Scientific Literature:**

Structure preservation is a major theme in classical numerical schemes, perhaps the authors can mention a few references (texts, survey papers).

**Theoretical Claims:**

The RTNN is constructed to theoretically satisfy the DSFT constraints, and the proof is straightforward. So numerically the fields satisfy the constraints to machine precision.

---

> ### Author Rebuttal · Authors · 2025-03-31
>
> We thank the reviewer for the detailed comments and suggestions. We appreciate them and hope to address some of the questions below:
>
> **Suggestions:**
> We agree that including representative solution plots would help the reader infer the difficulty in PINNs solution. We also appreciate the suggestion to cite relevant literature on structure-preserving numerical methods. Both additions will be included in the camera-ready version.
>
> **Why do RTNNs perform better than PINNs:**
>
> PINNs are notoriously difficult to optimize despite their recent success [1] . From the very advent of PINNs, several authors have advocated modifying machine learning models such that the boundary conditions or conservation laws are exactly zero. This has been empirically shown to improve training accuracy, including in our own work.  The work in **De Ryck et.al** [2] links the training dynamics of PINNs to the condition number of a Hermitian operator A that depend both on the PDE operator and the boundary operator. They show that hard-imposing constraints within the architecture reduces the condition number of A, which in turn improves the training dynamics and accelerates convergence.
>
> Another way of explaining the superior performance of RTNNs relative to unconstrained networks is their ability to capture and exploit the intrinsic relationships between physical fields. In many fluid dynamics problems, one must simultaneously model a scalar density field, a vector velocity field, and a scalar pressure field. Conventional unconstrained models typically treat these components as independent scalar output channels, thereby ignoring the important geometric and physical correlations—such as the interdependence between the directional components of the velocity field, or the influence of pressure gradients on both density and velocity. By contrast, RTNNs treat these quantities as a unified field coupled through a shared conservation-law-based flux, this in turn leads to superior accuracy during the training.
> We believe this discussion will improve the quality of our manuscript. We will include a dedicated paragraph on this topic in the final version, and we thank the reviewer for raising this point.
>
> **Question on Differentiation Layer:**
> In our implementation, we compute at most third-order derivatives, and all experiments were conducted in double precision (we will state this explicitly in the revised manuscript). While repeated differentiation can in principle lead to numerical issues, we did not observe any instabilities or degradation in constraint satisfaction. The resulting tensors remained divergence-free and symmetric to machine precision throughout. In fact, we also verified that this property holds under single precision as well (although naturally this leads to degradations in accuracy for all methods).
>
> We hope that we addressed the reviewer's questions and we will stay attentive for further questions throughout the review process.
>
>
> References:
>
> [1] Wang, S., Yu, X., & Perdikaris, P. (2020). When and why PINNs fail to train: A neural tangent kernel perspective. arXiv preprint arXiv:2007.14527.
>
> [2] De Ryck, T., Bonnet, F., Mishra, S., & de Bézenac, E. (2024). An Operator Preconditioning Perspective on Training in Physics-Informed Machine Learning. ICLR 2024.

---

### Official Review · Reviewer_yLyj · 2025-03-18

**Overall Recommendation:** 3

**Summary:**

The paper proposes RTNNs, which can encodes the divergence-free constraints in neural networks within the PINN framework. The divergence-free constraint is satisfied by computing the hessian matrix of a feed-forward MLP network and combining with a special-designed basis. The method is evaluated on a range of PDE tasks requiring the divergence-free condition.

**Claims And Evidence:**

- Although understanding the theorem in section 2.1 seems to be beyond my knowledge, the description of divergence-free construction using hessian matrix is clear and convincing.

- The experimental results look strong compared to PINN and NCL, although I am not familiar with the latter one.

**Essential References Not Discussed:**

- Not I am aware of.

**Experimental Designs Or Analyses:**

- Line 243 right: "Both methods use similar MLP architectures for fairness." Are they exactly the same?

- Is NCL supposed to be a strong baseline? The results of NCL in table 1 seems to be worse than PINN.

**Methods And Evaluation Criteria:**

- The benchmarked problems are reasonable and diverse.

- In line 270 left it says "$L_{data}$ incorporates supervised learning by penalizing the discrepancies between the model predictions and observed data labels". But in line 236 right it says "training is performed entirely without labeled data". It seems to be inconsistent.

**Other Comments Or Suggestions:**

- I did not notice typo.

**Other Strengths And Weaknesses:**

- The paper is well written.

- The method still requires other soft constraints to satisfy other constraints.

- The use neural network model is rather simple.

**Questions For Authors:**

- Will computing Hessian matrix affect the scaling of computation cost?

- What does the parentheses in equation 2 and 3 in subscripts mean?

- Is there any intuitive way to understand theorem 2.1? For example, what does it mean in terms of quantities on the simulation grid?

**Relation To Broader Scientific Literature:**

- They are adequately discussed.

**Theoretical Claims:**

- I did not check the correctness of math but the theory seems to be sound.

---

> ### Author Rebuttal · Authors · 2025-03-31
>
> We would like to thank the reviewer for their time and feedback. Below we address the reviewer's questions:
>
> **Label Data Inclusion**
>
> While we can incorporate labeled data to penalize discrepancies between predictions and observations, for this experiment we did not. We acknowledge the inconsistency between line 270 and line 236 and will correct it in the final version.
>
> **Both methods use similar MLP architectures for fairness. Are they exactly the same?**
> Both methods share identical hidden-layer configurations (depth, width, activation). The only difference is the *final output layer*, where each architecture produces the specific number of scalars required by its formulation.
>
> **Is NCL a Strong Baseline?**
> We address the NCL performance issue for the compressible case in our response to reviewer G2wL.
>
> **Will computing the Hessian matrix affect the scaling of computation cost?**
> A key contribution of our work is the formulation of a computationally tractable ansatz that scales well in the 2D+1 and 3D+1 cases of interest. Rather than naively parameterizing a (0,4)-tensor— which would require outputting \\( \mathcal{O}(N^4) \\) , where
> N is the number of spacial components components — we exploit an isomorphism with two-forms to reduce the number of scalar outputs to \\( \tfrac{N(N+1)}{2} \\), thereby limiting the number of second-order derivatives that must be computed.
>
> Computing the Hessian is inherently expensive; however, we address this challenge through two derivative evaluation strategies:
>
> - **Taylor-mode Automatic Differentiation:**
>   We compute second derivatives using Taylor-mode AD, which efficiently propagates truncated Taylor expansions ("jets"), thereby sharing repeated derivative computations across different orders through the network. For a scalar-output MLP, the complexity scales as \\( \mathcal{O}(L d W^2) \\), where \\( L \\) is the number of layers, \\( W \\) is the layer width, and \\( d \\) is the order of the derivative. This ensures that, for typical 2D+1 and 3D+1 PDEs, the cost of higher-order derivatives remains within a modest constant factor of a standard forward pass.
>
> - **SPINNs for Magneto-Hydrodynamics:**
>   In Section 3.4, we take advantage of the SPINN architecture[1] to decompose multi-dimensional derivative evaluations into per-axis components, reducing the cost from \\( \mathcal{O}(N^d) \\) to \\( \mathcal{O}(N\ d) \\), where \\( N \\) is the number of collocation points per axis and \\( d \\) is the order of the derivative.
>
> Together, these strategies ensure that while computing the Hessian is expensive, our approach remains scalable and computationally tractable for the 2D+time and 3D+time problems of interest.
>
> **What do the parentheses in equation 2 and 3 in the subscripts mean?**
> In tensor algebra, a pair of indices enclosed in round parentheses denotes symmetrization with respect to those indices. For example:
>
> \\[
> T_{(ab)\dots} = \frac{1}{2} \left( T_{ab\dots} + T_{ba\dots} \right)
> \\]
>
> **Intuition Behind Proof 2.1:**
>
>  At each grid point, we must generate a flux tensor that satisfies the DSFT condition exactly. Theorem 2.1 provides a systematic way to achieve this. We take advantage of the fact that the divergence of a second derivative of a four-tensor naturally vanishes under certain symmetries (2)--(4). Specifically, if we define $S_{ab}$ in terms of a second derivative of some four-tensor $K_{abcd}$, the divergence-free condition is automatically satisfied due to the symmetry properties of partial derivatives.
>
> This is reminiscent of how potentials are used in physics to impose constraints on derived quantities. For example, in electromagnetism, introducing a four-potential \\( A_\mu \\) ensures that the field strength tensor
>
> \\[
> F_{\mu\nu} = \partial_\mu A_\nu - \partial_\nu A_\mu
> \\]
>
> automatically satisfies the Bianchi identity:
>
> \\[
> \partial_\lambda F_{\mu\nu} + \partial_\nu F_{\lambda\mu} + \partial_\mu F_{\nu\lambda} = 0
> \\]
>
> We hope that we addressed the reviewer's questions sufficiently and we will stay attentive for further questions throughout the review process.
>
> **References :**
>
>
> [1] ​Cho, J., Nam, S., Yang, H., Yun, S.-B., Hong, Y., & Park, E. (2023). Separable Physics-Informed Neural Networks. Advances in Neural Information Processing Systems 36 (NeurIPS 2023).

---

### Decision · Program_Chairs · 2025-05-01

**Decision:**

Accept (poster)

**Comment:**

The paper proposes RTNNs, which can encodes the divergence-free constraints in neural networks within the PINN framework. The results show a clear advantage compared to standard PINNs. Some major concerns on experimental comparisons and evaluation metrics were successfully addressed by the authors' rebuttal.  These should be incorporated into the camera-ready version.